# Hopfion rings in a cubic chiral magnet

Fengshan Zheng[1,2,3✉], Nikolai S. Kiselev[3,4✉], Filipp N. Rybakov[5✉], Luyan Yang[6], Wen Shi[2,3], Stefan Blügel[3,4] & Rafal E. Dunin-Borkowski[2,3]

Magnetic skyrmions and hopfions are topological solitons[1]—well-localized field configurations that have gained considerable attention over the past decade owing to their unique particle-like properties, which make them promising objects for spintronic applications. Skyrmions[2,3] are two-dimensional solitons resembling vortex-like string structures that can penetrate an entire sample. Hopfions[4–9] are three-dimensional solitons confined within a magnetic sample volume and can be considered as closed twisted skyrmion strings that take the shape of a ring in the simplest case. Despite extensive research on magnetic skyrmions, the direct observation of magnetic hopfions is challenging[10] and has only been reported in a synthetic material[11]. Here we present direct observations of hopfions in crystals. In our experiment, we use transmission electron microscopy to observe hopfions forming coupled states with skyrmion strings in B20-type FeGe plates. We provide a protocol for nucleating such hopfion rings, which we verify using Lorentz imaging and electron holography. Our results are highly reproducible and in full agreement with micromagnetic simulations. We provide a unified skyrmion–hopfion homotopy classification and offer insight into the diversity of topological solitons in three-dimensional chiral magnets.

Topological magnetic solitons[1] are localized magnetic textures that have properties similar to those of ordinary particles. In particular, they can mutually interact and move under the influence of external stimuli. Chiral magnetic skyrmions[12–15] are notable examples of such objects. In thick three-dimensional (3D) samples, the magnetization vector field of skyrmions usually forms vortex-like strings that can penetrate an entire sample from one surface to another. Such filamentary magnetic textures, which arise as a consequence of competition between Heisenberg exchange and the Dzyaloshinskii–Moriya interaction (DMI)[16,17], have been observed in noncentrosymmetric crystals using various experimental techniques[2,3].

Recent studies have shown that skyrmion strings in isotropic chiral magnets are not rigid textures, but can twist and bend[18–21]. A prominent example of such elastic properties of skyrmion strings is the formation of skyrmion braids[20]—rope-like structures composed of skyrmion strings that wind around each other. Here we report a fundamentally different phenomenon. We use state-of-the-art transmission electron microscopy (TEM) and micromagnetic simulations to show that twisted skyrmion strings can be bent into rings in magnetic crystals, leading to the emergence of a different topology.

The concept of closed twisted skyrmion strings in field theories was first proposed by Ludwig Faddeev in 1975 (ref. 4). Such structures are now commonly termed hopfions[9], after Heinz Hopf, who built the theory that underlies their homotopy classification[22]. According to this theory, localized magnetization field configurations $\mathbf{m}(\mathbf{r}) = \mathbf{M}(\mathbf{r})/M_s$ can be classified according to the linkage of their fibres $\{\mathbf{m} = \mathbf{m}_P\}$, which, for any fixed points $P$ on a unit sphere $\mathbb{S}^2$, represent closed loops in $\mathbb{R}^3$

space. When any pair of such loops is linked once, as a pair of chain segments, the Hopf index of the field configuration is one, $H = 1$, and this configuration contains a hopfion.

Previous theoretical studies have predicted the existence of confined and isolated hopfions in chiral magnets[8,23–25]. However, the mechanism for hopfion stabilization that we describe below is essentially different. In our experiment, hopfions appear as rings around skyrmion strings and remain stable exclusively owing to intrinsic interactions and not as a result of the sample shape. The confined geometry of the sample has an essential role only in the nucleation of these hopfions, and is not relevant for their stability. To distinguish between hopfions that appear around skyrmion strings from isolated hopfions and hopfions in a nanoscale multilayer disk[11], we refer to the former as hopfion rings.

## Micromagnetic simulations of hopfion rings

Figure 1a–e illustrates the concept of hopfion rings and the process of their nucleation, according to the field-swapping protocol used in our experimental set-up below. In these micromagnetic simulations, we used a 0.5-μm-diameter disk with a thickness of 180 nm, with the external magnetic field applied parallel to the central axis of the disk. The magnetic states shown in Fig. 1a–e were obtained by direct energy minimization of the micromagnetic functional for an isotropic chiral magnet, taking into account demagnetizing fields. We also took into account the presence of a thin damaged layer on the sample surface (Fig. 1a), which typically results from sample preparation by focused ion beam (FIB) milling. (See Methods for more details about micromagnetic

[1]Spin-X Institute, Electron Microscopy Center, School of Physics and Optoelectronics, State Key Laboratory of Luminescent Materials and Devices, Guangdong-Hong Kong-Macao Joint Laboratory of Optoelectronic and Magnetic Functional Materials, South China University of Technology, Guangzhou, China. [2]Ernst Ruska-Centre for Microscopy and Spectroscopy with Electrons, Forschungszentrum Jülich, Jülich, Germany. [3]Peter Grünberg Institute, Forschungszentrum Jülich, Jülich, Germany. [4]Institute for Advanced Simulation, Forschungszentrum Jülich and JARA, Jülich, Germany. [5]Department of Physics and Astronomy, Uppsala University, Uppsala, Sweden. [6]Institute of Microstructure and Properties of Advanced Materials, Faculty of Materials and Manufacturing, Beijing University of Technology, Beijing, China. ✉e-mail: zhengfs@scut.edu.cn; n.kiselev@fz-juelich.de; philipp.rybakov@physics.uu.se

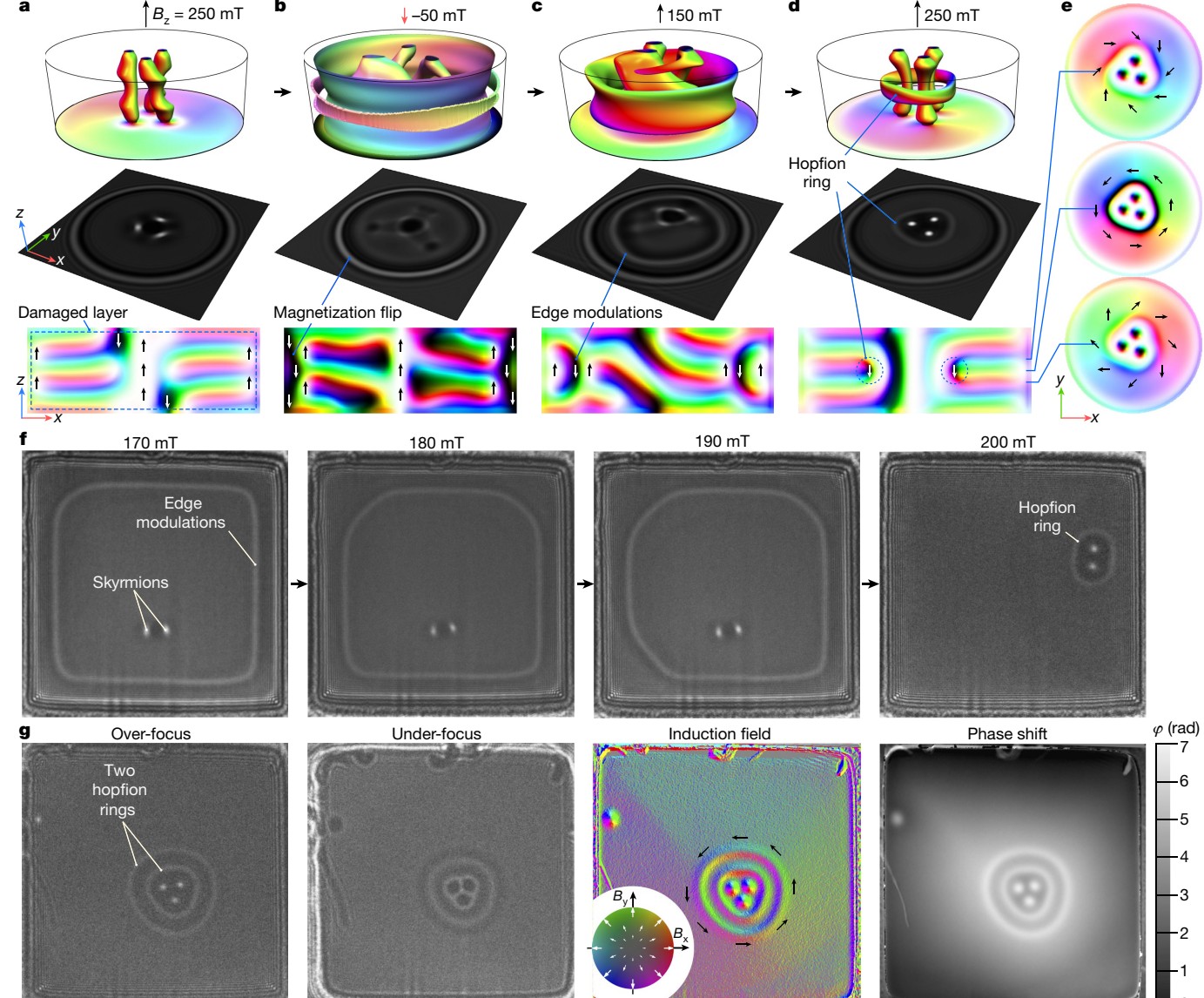

**Fig. 1 | Hopfion rings on skyrmion strings in FeGe samples of confined geometry.** Sequence of magnetic states in a 180-nm-thick, 0.5-μm-diameter disk obtained by energy minimization in different external magnetic fields applied perpendicular to the disk plane. **a**–**d**, Field-swapping protocol for the nucleation of a hopfion ring. **a**, The initial configuration. **b**, The state at a reversed field direction. **c**, The edge modulations formed at the magnetic field reversed back to a positive direction. **d**, The hopfion ring formed with increasing field. Top row, the magnetization on the lower face of the disk, as well as isosurfaces of $m_z = 0.5$ in **c** and $m_z = 0$ elsewhere. Middle row, over-focus Lorentz images of corresponding magnetic states calculated for a defocus distance of 400 μm with the electron beam parallel to the disk axis. Bottom row, the magnetization in a diametrical section of the sample. The dashed line in **a** marks the borders of a 7.5-nm-thick FIB-damaged layer (see Methods). **e**, The magnetization in transverse sections of the sample for the final state in **d**, with a hopfion ring on three skyrmion strings. The distance between the transverse sections is 10 nm. The colour code is explained by arrows superimposed on the diametrical and transverse sections: white and black correspond to $m_z = 1$ and $m_z = -1$, respectively; red–green–blue defines the direction of magnetization in the $xy$ plane. **f**, Experimental over-focus Lorentz images recorded in an FeGe plate of dimensions 1 μm × 1 μm and with a thickness of 180 nm. **g**, Experimental Lorentz images of double hopfion rings on three skyrmion strings at 200 mT and a corresponding magnetic induction map reconstructed from a phase-shift image recorded using off-axis electron holography. The experimental images in **f**,**g** were recorded at $T = 95$ K and at a defocus distance of 400 μm.

calculations, the properties of the damaged layer and Lorentz image simulations.)

Figure 1a shows an initial state with three skyrmion strings, demonstrating a weak braiding effect and conical modulations along the sample thickness. In a confined geometry, such modulations exhibit an additional twist that resembles a vortex-like structure[26,27] in any transverse section. Swapping the direction of the external magnetic field leads to a flip of the magnetization along the perimeter of the disk, as shown in Fig. 1b. To prevent the collapse of the skyrmions in a negative field, we apply a relatively weak field of −50 mT. Swapping the field back to the positive direction results in the appearance of edge modulations, representing a volume with magnetization pointing against the external field (Fig. 1c). On further increasing the field in the positive direction, the edge modulations contract towards the centre of the sample, forming a hopfion ring around three skyrmion strings (Fig. 1d). Figure 1d shows that each colour line (fibre) winds about the isosurface of the hopfion ring only once, which indicates that the Hopf index of the corresponding texture is nontrivial. The middle cross-section of the hopfion ring surrounding the three skyrmion strings shown in Fig. 1e resembles a skyrmion bag, as predicted

theoretically for a two-dimensional (2D) model of a chiral magnet[28,29]. Our analysis shows that such textures have negative skyrmion topological charge, $Q$, in all transverse sections (see Methods). However, the rings shown in Fig. 1d represent hopfions, which is not possible in 2D materials[30] and is different from skyrmion bags with a positive topological charge[31]. It is therefore essential to distinguish between magnetic textures with hopfion rings and skyrmion bags.

## Experimental observation of hopfion rings

Following the protocol suggested by the micromagnetic simulations, we demonstrated the nucleation of hopfion rings in a square-shaped FeGe sample of lateral size 1 μm × 1 μm and thickness 180 nm. (See Methods for details about sample preparation and the TEM experimental set-up.) A representative example of a magnetic configuration with edge modulations after field swapping is shown in Fig. 1f. On increasing the external field, the bright contrast loop of the edge modulations contracts around two skyrmions. The final state (at 200 mT) corresponds to a hopfion ring around two skyrmion strings. The intermediate configurations at 180 mT and 190 mT are seldom observed in our experiments. In most cases, contraction of the edge modulations occurs abruptly on a timescale that is much shorter than the temporal resolution of the detector in our TEM.

As a result of the excitation of the magnetization induced by the abrupt contraction of the edge modulations and thermal fluctuations, the above protocol has a probabilistic character. The final state shown in Fig. 1f (at 200 mT) is an example of a successful contraction.

Notably, the sequential application of the above field-swapping protocol can result in the nucleation of a second hopfion ring with high probability. Figure 1g shows an example of such a double hopfion ring around three skyrmion strings. Further images with multiple hopfion rings are provided in Extended Data Figs. 5 and 6. These images illustrate the successful contraction of the second ring. However, because hopfion rings are metastable states, further application of the field-swapping protocol can cause a transition to a lower energy state, resulting in the collapse of the hopfion rings.

The above protocol becomes more reliable at a higher sample temperature, indicating that thermal fluctuations have a key role in hopfion-ring nucleation. However, the range of applied magnetic fields over which the hopfion rings remain stable decreases gradually with increasing temperature. We found an optimal range of $T = 180–200$ K, over which the above protocol exhibits the greatest efficiency. Below 180 K, nucleation of hopfion rings is still possible, but typically requires more field-swapping cycles. At a lower temperature, the edge modulations can move towards the edges of the sample and disappear. By contrast, at higher temperatures the edge modulations can contract towards the centre of the sample. Above 200 K, abrupt contraction of the edge modulations can lead to their collapse. The behaviour of the edge modulations at different temperatures can be explained by the presence of an energy barrier that prevents their contraction towards the centre of the domain. The probability of overcoming this energy barrier then becomes greater at high temperature.

The process of hopfion-ring nucleation is shown in Supplementary Videos 1–5. These videos were captured in situ at a temperature of $T = 180$ K. We performed several field-swapping cycles in the first stage with a small amplitude of approximately ±50 mT. This step was designed to generate edge modulations that formed closed loops and propagated towards the centre of the sample. Once one or a few of these loops had been created, we gradually increased the applied magnetic field up to approximately 150 mT, which resulted in the formation of various hopfion rings.

The tilt angle of the external magnetic field to the plate normal is an essential parameter for hopfion-ring nucleation. In our experiment, we found that the tilt angle of the field should not exceed 5°. Otherwise, the edge modulations mainly form on one side of the sample, resulting in a strongly asymmetric configuration.

## Diversity and in-field evolution of hopfion rings

Figure 2 illustrates the evolution of diverse configurations of hopfion rings in increasing applied magnetic fields. It should be noted that the Lorentz TEM images shown in Fig. 2 were recorded with the sample at different temperatures, which can affect the stability range of the hopfion rings and skyrmions. As a result, the applied magnetic field is only indicated schematically above the figure. The complete series of images, showing the entire field of view and exact values of the external magnetic field and specimen temperature, is provided in Extended Data Figs. 1, 2 and 3. Other representative images of multiple skyrmions surrounded by a single hopfion ring are provided in Extended Data Fig. 4. Additional Lorentz TEM images illustrating the in-field evolution of magnetic states with different topological charges and symmetries composed of skyrmions and hopfion rings are shown in Extended Data Figs. 5 and 6.

Figure 2 shows that the hopfion ring shrinks with increasing applied field. As a result, the distance between the skyrmions inside the ring decreases. By contrast, in ordinary skyrmion clusters (without a hopfion ring) the distance between the skyrmions increases with applied field[32]. Figure 2 shows that the symmetry of the magnetic texture of skyrmions and hopfion rings also changes with increasing applied field. For instance, the hopfion ring shown in the bottom row has a triangular shape at low field. With increasing magnetic field, it adopts a pentagonal and then a circular shape. Such symmetry transitions are found to be reversible with respect to increasing and decreasing field.

With increasing field, the contrast of the hopfion rings in Lorentz TEM images becomes weaker, suggesting that the magnetic modulations become localized in a smaller volume. The distance between the hopfion ring and the skyrmions also decreases, with bright spots of skyrmion strings nearly touching the hopfion ring just before it collapses. After the collapse of the hopfion ring, the distance between the skyrmions increases abruptly. In most cases, bright spots of weak contrast, which are identified as chiral bobbers[33] or dipole strings[34], are often observed. Their positions are marked by dashed circles in Fig. 2. Because chiral bobbers and dipole strings have nearly identical contrast in Lorentz TEM images[35], they cannot be distinguished reliably in these experiments. They disappear on increasing the applied magnetic field further. The appearance of such objects, which contain magnetic singularities, indicates that the collapse of a hopfion ring represents a topological transition.

Figure 3 shows exotic states with negative and positive topological charges obtained using the above protocol in a 180-nm-thick sample. Magnetic textures with such contrast in our experiments are observed less frequently than those depicted in Fig. 2 (see also Extended Data Fig. 5). The figure shows that a hopfion ring surrounding skyrmion strings can have different sizes, which seem to be limited only by the geometry of the sample. The latter observation agrees with theoretical predictions for 2D skyrmion bags[28], which also have no limitation in their size. The theoretical Lorentz TEM images shown in Fig. 3 are in excellent agreement with the experimental images.

Figure 4 and Extended Data Figs. 7 and 8 show corresponding magnetic textures obtained by direct energy minimization of the micromagnetic functional. In particular, Fig. 4 shows the 3D magnetic textures of the first four states in Fig. 3a–d, and Extended Data Fig. 8 shows magnetic textures of the other four states in Fig. 3e-h. By starting from different initial states, we found slightly different stable configurations for most of the configurations shown in Fig. 3. Figure 4 shows the magnetic textures with the lowest energies.

The hopfion rings shown in Fig. 4a,b,d are located in the middle plane of the sample. However, this is not always the case. In Fig. 4c, the hopfion ring is shifted slightly towards one of the surfaces and is also stretched slightly along the $z$ axis. A similar configuration is shown in Extended Data Fig. 8a. In these cases, the projected magnetization of the hopfion ring onto the $xy$ plane overlaps with the magnetization

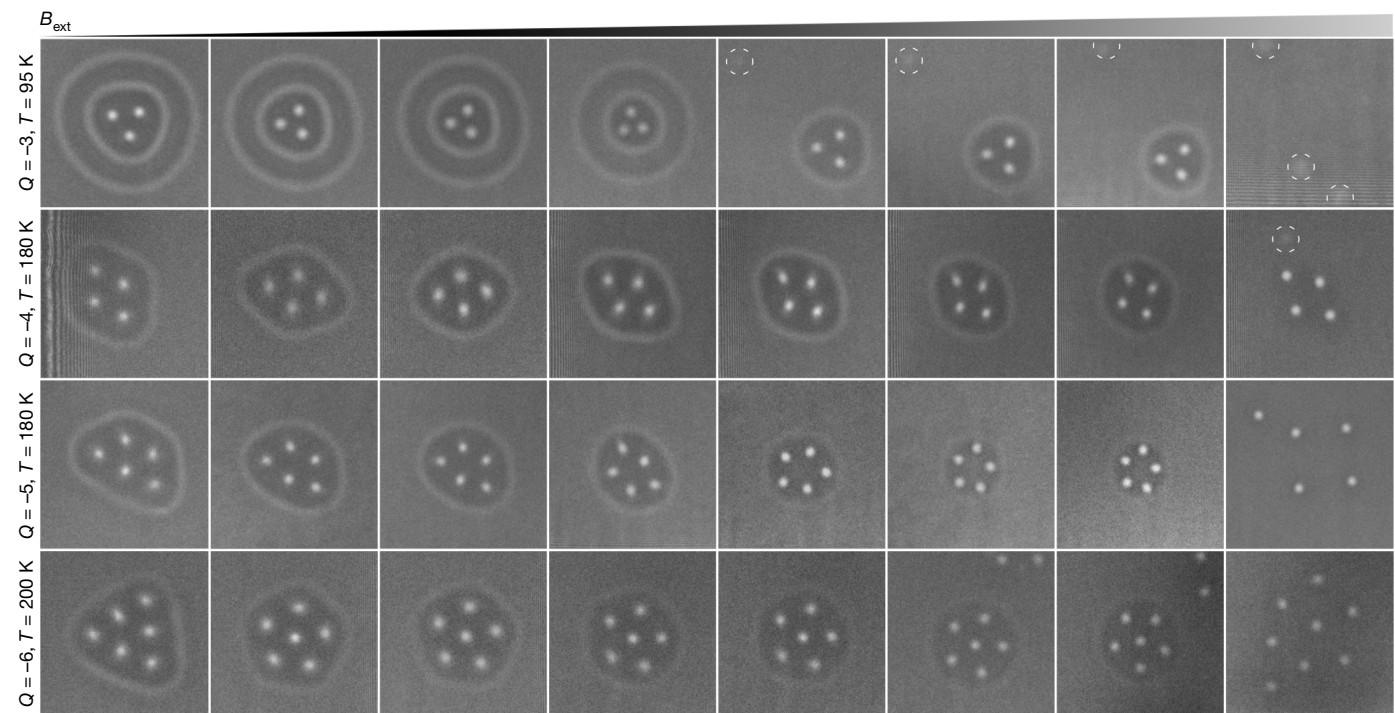

**Fig. 2 | Evolution of hopfion rings with an increasing applied magnetic field.** Each row shows the evolution of hopfion rings surrounding skyrmion strings, shown as a function of the external magnetic field from left to right in the form of experimental over-focus Lorentz images. The topological charge $Q$ and the temperature at which each row was recorded are indicated on the left side. The value of the perpendicular field increases from left to right between around 150 mT and around 450 mT. The values of the external magnetic field are given in Extended Data Figs. 1–3. The images have identical sizes of around $450 \times 450$ nm² and have been extracted from larger fields of view. The dashed circles in the images for $Q = -3$ and $Q = -4$ mark spots of low contrast, which result from the nucleation of chiral bobbers or dipole strings during the collapse of the hopfion ring.

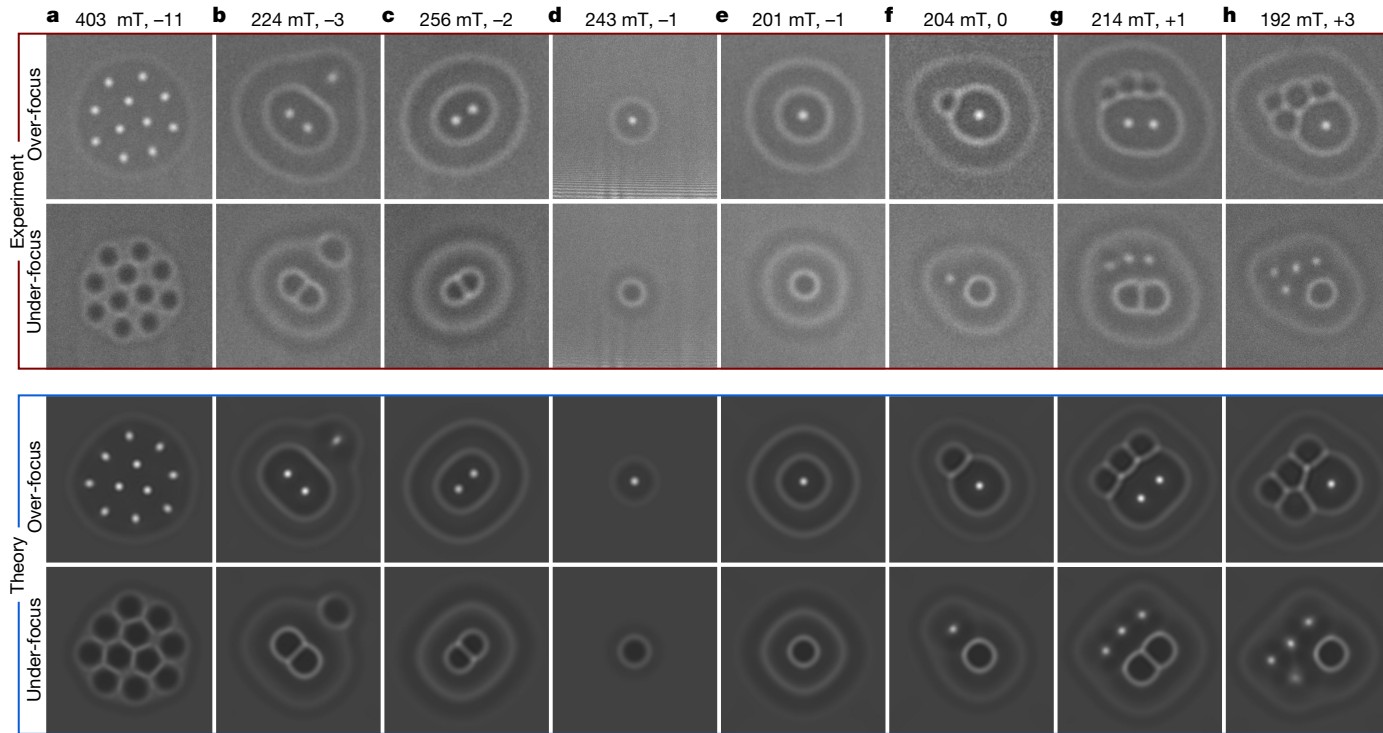

**Fig. 3 | Exotic magnetic states with hopfion rings. a–h,** Top, experimental Lorentz TEM images of magnetic states with different topological charges ($Q = -11$ (**a**), $Q = -3$ (**b**), $Q = -2$ (**c**), $Q = -1$ (**d,e**), $Q = 0$ (**f**), $Q = 1$ (**g**) and $Q = 3$ (**h**)) in different external magnetic fields in a 180-nm-thick sample recorded in the over-focus and under-focus regimes. The images in **c, f, h** were recorded at $T = 95$ K; the other images were recorded at $T = 180$ K. Bottom, theoretical Lorentz TEM images calculated for corresponding magnetic textures using micromagnetic simulations for identical external magnetic fields to the experimental values. All of the experimental and theoretical images have identical sizes of around $450 \times 450$ nm². The magnetization of the entire $1\,\mu m \times 1\,\mu m$ domain and corresponding theoretical Lorentz TEM images are provided in Extended Data Fig. 7. The 3D magnetization vector fields of the magnetic states shown in **a–d** are provided in Fig. 4. See Extended Data Fig. 8 for the 3D magnetization vector fields of the magnetic states shown in **e–h**.

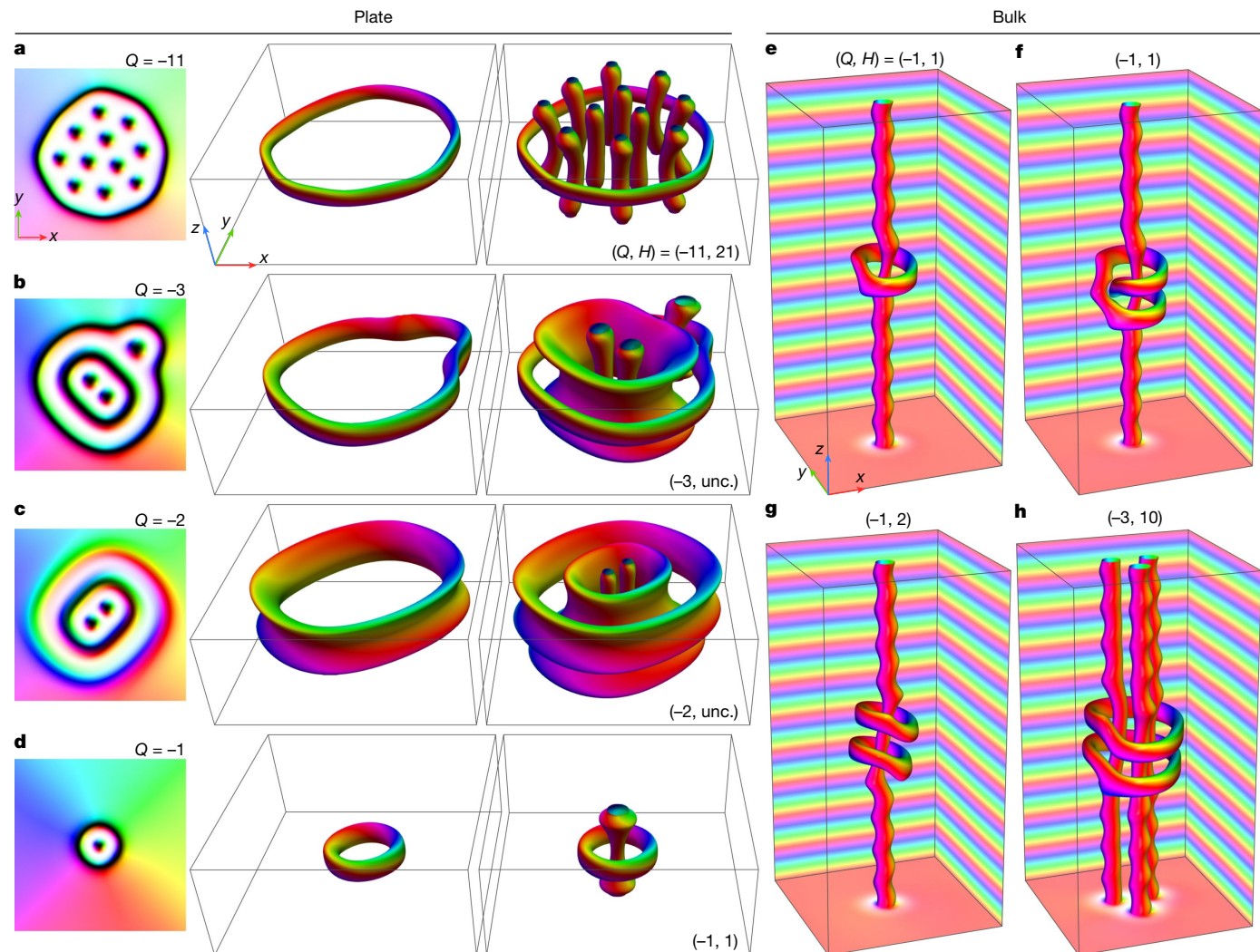

**Fig. 4 | Micromagnetic simulations of magnetic textures with hopfion rings. a–d,** Each row of images corresponds to the magnetic textures shown in Fig. 3a–d, with $Q = -11$ (**a**), $Q = -3$ (**b**), $Q = -2$ (**c**) and $Q = -1$ (**d**). The first column shows the magnetization field in the middle plane of the sample. The isosurfaces $m_z = 0$ shown in the second and third columns are identical and differ only in their representations. In particular, the second column shows the isosurface of the hopfion ring alone, to highlight the linkage of the pre-images. The third column shows complete isosurfaces of hopfion rings and skyrmions. Analogous images for the magnetic textures depicted in Fig. 3e–h are provided in Extended Data Fig. 8a–d. **e–h,** Images showing hopfion rings in a bulk system. As well as the isosurfaces $m_z = 0$, we show the magnetization at the edges of the simulated box to visualize the conical modulations along the $z$ axis (see also Extended Data Fig. 8e-h for other hopfion rings in the bulk). A pair of indices $(Q, H)$ defines the skyrmion–hopfion topological charge of the corresponding texture (see Methods). **e** and **f** show topologically identical states of different morphology. **g,** The pair of hopfion rings on a single skyrmion string. **h,** The pair of hopfion rings on three skyrmion strings. Note that, for the states shown in **b,c**, the Hopf charge is uncertain.

of the interior ring, leading to a visible contrast difference in Lorentz TEM images of the corresponding magnetic textures. For example, the outer rings shown in Fig. 4c,e have higher contrast when compared to the other images. The contrast in the theoretical images shows the same features. Additional comparisons between experimental and theoretical Lorentz TEM images and electron optical phase images are provided in Extended Data Figs. 9 and 10.

## Skyrmion–hopfion topological charge

It is essential to find a complete topological classification of the observed magnetic textures. Our homotopy-group analysis (see Methods) reveals that these textures are classified according to their skyrmion–hopfion topological charge, which can be represented by an ordered pair of integers, $(Q, H)$. Notably, the Hopf charge $H$ here depends not only on the intrinsic structure of the hopfion rings but also on their linking to skyrmion strings. For instance, for the configuration shown in Fig. 4d (right), if there were no skyrmion string at the

centre, the Hopf charge would be −1. However, linkage to the skyrmion string results in an actual Hopf charge of +1. Consequently, when such a ring detaches continuously from the string, there must be a profound transformation of its internal structure, leading to its conversion into an anti-hopfion (see Supplementary Video 6).

Despite the presence of hopfion rings in all of the magnetic textures shown in Fig. 4, it is worth emphasizing that we do not specify the Hopf charge for the cases depicted in Fig. 4b,c. Because the inner rings are near the open boundaries of the plate, the compactification condition is not satisfied (see Methods), and a homotopy classification based on the Hopf charge is unsuitable in this case. Another type of texture that does not satisfy the compactification condition is fractional hopfions[24,36].

## Discussion

In our experiments, the sizes of the hopfion rings are much smaller than the in-plane sample sizes, and hopfion rings can move in at least two spatial dimensions. In thicker samples, we speculate that hopfion

rings can also move and interact with each other in the third spatial dimension—along the skyrmion strings. Figure 4e–h and Extended Data Fig. 8e–h illustrate a wide diversity of stable solutions for hopfion rings in bulk systems (see Methods). On the basis of the general principles of classical field theory, it is understood that the Hopf charge of skyrmion strings can be affected by longitudinal twists of skyrmions, as well as by skyrmion braiding[37–39]. However, owing to the chiral nature of the DMI present in the system studied here, stable states with skyrmion twists of multiples of $2\pi$ are not observed. Nevertheless, we speculate that such states might be possible in systems that have frustrated exchange interactions[9,40]. On the other hand, the phenomenon of skyrmion braiding has already been demonstrated in chiral magnets[20]. Extended Data Fig. 8h shows an example of a skyrmion braid with two hopfion rings and $H = 12$. This example can be compared with Fig. 4h, which shows straight skyrmion strings surrounded by two hopfion rings and $H = 10$.

A notable property of the hopfion solutions is the presence of a zero mode arising from the intrinsic symmetry of the Hamiltonian (see Methods). The rotational–translational motion of the hopfion ring along the string preserves energy (see Supplementary Videos 7 and 8), which suggests high mobility of the hopfion rings.

Our findings open broad perspectives for the study of the dynamical[41,42] and transport properties[43,44] of hopfion rings, and will have practical applications in spintronics, neuromorphic computing and other technologies.

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

## Methods

### Micromagnetic simulations

The micromagnetic approach was followed in this work. The total energy of the system includes the exchange energy, the DMI energy, the Zeeman energy and the energy of the demagnetizing fields[45]:

$$\mathcal{E} = \int_{V_m} d\mathbf{r} \mathcal{A} \sum_{i=x,y,z} |\nabla m_i|^2 + \mathcal{D} \, \mathbf{m} \cdot (\nabla \times \mathbf{m}) - M_s \, \mathbf{m} \cdot \mathbf{B} + \tag{1}$$
$$+ \frac{1}{2\mu_0} \int_{\mathbb{R}^3} d\mathbf{r} \sum_{i=x,y,z} |\nabla A_{d,i}|^2,$$

where $\mathbf{m}(\mathbf{r}) = \mathbf{M}(\mathbf{r})/M_s$ is a unit vector field that defines the direction of the magnetization, $M_s = |\mathbf{M}(\mathbf{r})|$ is the saturation magnetization and $\mu_0$ is the vacuum permeability ($\mu_0 \approx 1.257 \, \mu\text{N A}^{-2}$). The constants $\mathcal{A}$ and $\mathcal{D}$ are the exchange stiffness and the isotropic bulk DMI, respectively. The ratio between $\mathcal{A}$ and $\mathcal{D}$ defines the equilibrium period of the conical phase, $L_D = 4\pi \mathcal{A}/\mathcal{D}$. The magnetic field in equation (1), $\mathbf{B} = \mathbf{B}_{\text{ext}} + \nabla \times \mathbf{A}_d$, is the sum of the external magnetic field and the demagnetizing field, where $\mathbf{A}_d(\mathbf{r})$ is the component of the magnetic vector potential induced by the magnetization. For the calculations in the bulk system, we set the external magnetic field $\mathbf{B}_{\text{ext}} = 0.5 \, B_D \hat{\mathbf{e}}_z$, where $B_D = \mathcal{D}^2/(2M_s \mathcal{A})$ is the conical phase saturation field in the absence of demagnetizing fields[20]. We used the following material parameters for FeGe[20,33]: $\mathcal{A} = 4.75 \, \text{pJ m}^{-1}$, $\mathcal{D} = 0.853 \, \text{mJ m}^{-2}$ and $M_s = 384 \, \text{kA m}^{-1}$. For the 0.5-μm-diameter and 180-nm-thick disk sample depicted in Fig. 1a–e, the calculations were performed on a mesh with $256 \times 256 \times 64$ cuboids. Calculations for the $1 \, \mu\text{m} \times 1 \, \mu\text{m} \times 180 \, \text{nm}$ sample were performed on a mesh with $400 \times 400 \times 72$ cuboids. For the bulk magnet, we exclude dipole–dipole interactions and consider a domain of size $5L_D \times 5L_D \times 10L_D$ under periodic boundary conditions on a mesh with $256 \times 256 \times 512$ cuboids.

Following the arguments presented in a previous study[35], a thin surface layer of the isotropic chiral magnet crystal is damaged during FIB milling and can be effectively approximated by material parameters that are identical to those of the bulk crystal, but with the DMI coupling constant set to zero. In the previous report[35], the thickness of the FIB-damaged layer of an FeGe nanocylinder was estimated to be $6 \pm 1 \, \text{nm}$. According to another report[21], the thickness of the damaged layer of an FeGe needle-like sample is around 10 nm. Here, we assume an intermediate thickness for the damaged layer of 7.5 nm (corresponding to three surface cuboids).

It should be noted that the presence or absence of a damaged layer in our simulations has almost no effect on the stability of the solutions shown in Fig. 4. The contrast in theoretical Lorentz TEM images in Fig. 3 also does not change significantly when the presence of a damaged layer is ignored. However, the presence of a damaged surface layer has a crucial role in hopfion-ring nucleation. In the simulations, through the application of a magnetic field in the negative and positive directions with respect to the $z$ axis, we only succeeded in observing hopfion-ring nucleation, as shown in Fig. 1a–d, in the presence of a damaged surface layer.

Statically stable solutions of the Hamiltonian (equation (1)) were found by using the numerical energy minimization method described previously[20] using the Excalibur code[46]. The solutions were double-checked using the publicly available software Mumax[47]. In the Supplementary information, we also provide three Mumax scripts, which can be used to reproduce the results of our micromagnetic simulations. Script I allows the hopfion-ring nucleation depicted in Fig. 1a–d to be reproduced. Because the states depicted in Fig. 1a,d are two states with different energies that are stabilized in identical conditions, the transition between them requires additional energy pumping. The energy balance between these states depends on the applied field. In the experimental set-up, this in-field transition is enhanced by thermal fluctuation and, as a result, has a probabilistic character. To make the nucleation of the hopfion ring deterministic (reproducible), in the micromagnetic simulations we use an abrupt switch of the magnetic field (with a step of around 100 mT) to overcome the barrier between the metastable states. Script II, with minor modifications of the initial states discussed in the next section, can be used to reproduce the states shown in Fig. 4. For a description of Script III, see the following section.

### Initial state for hopfion rings in micromagnetic simulations

On the basis of experimental observations and theoretical analysis, we noticed that the presence of the conical phase around different localized states results in an additional contribution to the electron optical phase shift that changes around the perimeter of the sample. To obtain the magnetic textures in nanoscale samples, we used initial configurations corresponding to a superposition of cylindrical domains, with their magnetization pointing up and down, embedded in a conical phase and with an additional phase modulation resembling a vortex in the $xy$ plane of the form

$$\Theta = \text{acos}\left(\frac{B_{\text{ext}}}{B_D + \mu_0 M_s}\right), \quad \Phi = \text{atan}\frac{y}{x} + \frac{\pi}{2} + kz, \tag{2}$$

where $k = 2\pi/L_D$ is the wave number. In another study[27], similar vortex-cone configurations were discussed in the context of screw dislocations in bulk chiral magnets. Here, a magnetic configuration approximated by equation (2) appears owing to an interplay between short-range interactions (Heisenberg exchange and DMI) and a long-range demagnetizing field. This effect has previously been observed in samples of confined geometry[26,32,33].

Representative examples of two initial states are illustrated in Extended Data Fig. 9a. Stable magnetic states obtained from these initial states after energy minimization are shown in Extended Data Fig. 9b,c. The state with a compact hopfion ring not only has lower energy, but also provides contrast in theoretical Lorentz TEM images that accurately match experimental images (Extended Data Fig. 9e,f). The results shown in Extended Data Fig. 9 can be reproduced by using Mumax Script II.

For simulations of the bulk, skyrmion strings with hopfion rings were embedded into the uniform conical phase:

$$\Theta = \text{acos}(B_{\text{ext}}/B_D), \quad \Phi = kz. \tag{3}$$

For hopfion rings, we used the following toroidal ansatz:

$$\Theta = \pi\left(1 - \frac{\eta}{R_1}\right), \quad 0 \le \eta \le R_1, \tag{4}$$

$$\Phi = \text{atan}\left(\frac{y}{x}\right) - \text{atan}\left(\frac{z}{R_2 - \rho}\right) - \frac{\pi}{2}, \tag{5}$$

where $R_1$ and $R_2$ are the minor and major radii of a torus, respectively, $\rho = \sqrt{x^2 + y^2}$ and $\eta = \sqrt{(R_2 - \rho)^2 + z^2}$. We also refer the reader to the Mumax Script III for initial state implementation. By default, Script III can reproduce a complex configuration in the bulk system, as shown in Extended Data Fig. 8h. With minor modifications, it can also be used to replicate all other states.

### Simulations of electron optical phase-shift and Lorentz TEM images

By using the phase object approximation and assuming that the electron beam is antiparallel to the $z$ axis, the wave function of an electron beam can be written as follows[48]:

$$\Psi_0(x, y) \propto \exp(i\varphi(x, y)),\tag{6}$$

where $\varphi(x, y)$ is the magnetic contribution to the phase shift[49]

$$\varphi(x, y) = \frac{2\pi e}{h} \int\limits_{-\infty}^{+\infty} dz \, \mathbf{A}_d \cdot \hat{\mathbf{e}}_z,\tag{7}$$

$e$ is an elementary (positive) charge (around $1.6 \times 10^{-19}$ C) and $h$ is Planck's constant (approximately $6.63 \times 10^{-34}$ m$^2$ kg s$^{-1}$). Because our approach for the solution of the micromagnetic problem recovers the magnetic vector potential $\mathbf{A}_d$, simulation of the electron optical phase shift is straightforward.

In the Fresnel mode of Lorentz TEM, neglecting aberrations other than defocus, aperture functions and sources of incoherence and blurring, the wave function at the detector plane can be written in the form

$$\Psi_{\Delta z}(x, y) \propto \int \int dx'dy' \, \Psi_0(x', y')K(x - x', y - y'),\tag{8}$$

where the kernel is given by the expression

$$K(\xi, \eta) = \exp\left(\frac{i\pi}{\lambda \Delta z}(\xi^2 + \eta^2)\right),\tag{9}$$

the relativistic electron wavelength is

$$\lambda = \frac{hc}{\sqrt{(eU)^2 + 2eUm_ec^2}},\tag{10}$$

$\Delta z$ is the defocus of the imaging lens, $c$ is the speed of light (approximately $2.99 \times 10^8$ m s$^{-1}$), $U$ is the microscope accelerating voltage and $m_e$ is the electron rest mass (around $9.11 \times 10^{-31}$ kg). The image intensity is then calculated using the expression

$$I(x, y) \propto |\Psi_{\Delta z}(x, y)|^2.\tag{11}$$

For more details about the calculation of Lorentz TEM images, see ref. 20.

### Homotopy-group analysis

**Skyrmion topological charge.** For magnetic textures localized in the plane area $\Omega \subseteq \mathbb{R}^2$, such that at the boundary of this area $\partial\Omega$ the magnetization field $\mathbf{m}(\partial\Omega) = \mathbf{m}_0$, the classifying group is the second homotopy group of the space $\mathbb{S}^2$ at the base point $\mathbf{m}_0$, and there is an isomorphism to the group of integers (Abelian group with respect to addition):

$$\pi_2(\mathbb{S}^2, \mathbf{m}_0) = \mathbb{Z}.\tag{12}$$

This implies that any continuous magnetic texture satisfying the above criteria of localization can be attributed to an integer number, which is commonly referred to as the skyrmion topological charge (or skyrmion topological index), and can be calculated as follows:

$$\begin{cases} Q = \dfrac{1}{4\pi} \displaystyle\int_\Omega dr_1 dr_2 \, \mathbf{F} \cdot \hat{\mathbf{e}}_{r_3}, \\ \mathbf{m}_0 \cdot \hat{\mathbf{e}}_{r_3} > 0, \end{cases}\tag{13}$$

where

$$\mathbf{F} = \begin{pmatrix} \mathbf{m} \cdot [\partial_{r_2}\mathbf{m} \times \partial_{r_3}\mathbf{m}] \\ \mathbf{m} \cdot [\partial_{r_3}\mathbf{m} \times \partial_{r_1}\mathbf{m}] \\ \mathbf{m} \cdot [\partial_{r_1}\mathbf{m} \times \partial_{r_2}\mathbf{m}] \end{pmatrix}\tag{14}$$

is the vector of curvature[50,51], which is also known (up to a prefactor) as the gyro-vector or vorticity[10,52–54], and $r_1$, $r_2$ and $r_3$ are local right-handed Cartesian coordinates.

The unit field $\mathbf{m}$ can be parameterized on the $\mathbb{S}^2$ sphere using polar and azimuthal angles $\Theta$ and $\Phi$, respectively, in the form $\mathbf{m} = (\cos\Phi\sin\Theta, \sin\Phi\sin\Theta, \cos\Theta)$. The corresponding topological invariant, up to the sign, is the degree of mapping of the skyrmion localization area onto the sphere[55], which can be calculated using the top part of equation (13), assuming that $r_1$ and $r_2$ lie in the skyrmion plane. It should be noted that the sign of the integral in the top part of equation (13) depends on the choice of the orientation of the coordinate system. For example, in Fig. 4a the sign of $Q$ depends on whether the $r_3$ axis is parallel or antiparallel to the $z$ axis and equals $-11$ or $11$, respectively. The condition in the bottom part of equation (13) removes this ambiguity. A justification for this statement, based on the theory of fundamental invariants, can be found in a previous study[56]. The local coordinate system ($r_1$, $r_2$, $r_3$) for calculating the topological charge $Q$ of a particular skyrmion is chosen according to the condition in the bottom part of equation (13).

For skyrmions that have different $\mathbf{m}_0$ in the global coordinate system, equation (12) is not globally applicable because the base points[57] are different. However, isomorphisms to the group of integers can always be completed through continuous individual transformations of vector fields to match the vectors $\mathbf{m}_0$ to one base point.

Here, we use the same convention for the sign of the topological charge as previous reports[20,28,29,58], such that an elementary Bloch-type or Neel-type skyrmion has $Q = -1$.

**Hopfion topological charge.** For a magnetic texture localized within the 3D domain $\Omega \subseteq \mathbb{R}^3$, with a fixed magnetization $\mathbf{m}(\partial\Omega) = \mathbf{m}_0$ at the boundary $\partial\Omega$ of the domain, the classifying group corresponds to the third homotopy group of the space $\mathbb{S}^2$ at the base point $\mathbf{m}_0$:

$$\pi_3(\mathbb{S}^2, \mathbf{m}_0) = \mathbb{Z}.\tag{15}$$

The corresponding topological charge, which is known as the Hopf invariant, can be calculated using Whitehead's formula[51,59]:

$$H = -\frac{1}{16\pi^2} \int_\Omega dr_1 dr_2 dr_3 \, \mathbf{F} \cdot [(\nabla\times)^{-1}\mathbf{F}].\tag{16}$$

**Skyrmion–hopfion topological charge.** To analyse the continuous texture localized on a segment of a skyrmion string, we use the compactification approach and other methods of algebraic topology[60]. First, we note that, owing to the invariance of $Q$ along a skyrmion string, the lower and upper cross-sections bounding the skyrmion string segment are related by a trivial transformation. This implies that the dimension along the skyrmion string can be compactified to a circle $\mathbb{S}^1$. Second, we note that the conical phase and the phase with uniform magnetization $\mathbf{m}_0$ are equivalent to each other up to a trivial transformation. By exploiting this observation, one can compactify the remaining two dimensions. The magnetic texture localized on a segment of a skyrmion string can be treated as if it is confined within a solid torus $\Omega = \mathbb{D}^2 \times \mathbb{S}^1$. The noncollinearities of $\mathbf{m}$ are then localized inside $\Omega$, while everywhere on its surface $\partial\Omega = \mathbb{S}^1 \times \mathbb{S}^1$ the magnetization $\mathbf{m}(\partial\Omega) = \mathbf{m}_0$ is fixed.

Thereby, the homotopy classification arises from a continuous map from a one-point compactified solid torus to the spin space:

$$\mathbb{D}^2 \times \mathbb{S}^1/\mathbb{S}^1 \times \mathbb{S}^1 \to \mathbb{S}^2.\tag{17}$$

By using the homeomorphism of the quotient spaces $\mathbb{D}^2 \times \mathbb{S}^1/\mathbb{S}^1 \times \mathbb{S}^1$ and $\mathbb{S}^3/\mathbb{S}^1$, as well as the homotopy equivalence between $\mathbb{S}^3/\mathbb{S}^1$ and $\mathbb{S}^2 \vee \mathbb{S}^3$, we find a homotopy equivalent map

$$\mathbb{S}^2 \vee \mathbb{S}^3 \to \mathbb{S}^2. \tag{18}$$

Taking the base point, $\mathbf{m}_0$, as a point common to the wedge sum, we immediately find the homotopy group

$$G = \pi_2(\mathbb{S}^2, \mathbf{m}_0) \times \pi_3(\mathbb{S}^2, \mathbf{m}_0) = \mathbb{Z} \times \mathbb{Z}, \tag{19}$$

where $\pi_2$ and $\pi_3$ correspond to equations (12) and (15), respectively, and the components of the topological charge are subject to equations (13) and (16), respectively. The topological index for the textures depicted in Fig. 4 and Extended Data Fig. 8 then represents the ordered pair of integers $(Q, H)$.

## Calculation of topological charges

To compactify the textures obtained in micromagnetic simulations, we used the nested box approach. This method involves fixing and placing the box containing the studied texture at the centre of a slightly larger computational box. The computational box has periodic boundary conditions along the $z$ direction, and the remaining boundaries are fixed $\mathbf{m}_0 = \hat{\mathbf{e}}_z$. To ensure continuity of the vector field $\mathbf{m}$ in the transition regions between the nested boxes, we minimized the Dirichlet energy, $\int d\mathbf{r} |\nabla \mathbf{m}|^2$. Next, to calculate $\mathbf{F}$ and the topological charge $Q$, we used a previously proposed lattice approach[61]. The vector potential of the divergence-free field $\mathbf{F}$ was obtained by evaluating the integral:

$$(\nabla \times)^{-1}\mathbf{F} = \int dx\ \mathbf{F} \times \hat{\mathbf{e}}_x. \tag{20}$$

The Hopf index $H$ was then determined by numerically integrating equation (16).

For additional verification, we also computed the index $H$ by calculating the linking number for curves in real space that corresponded to two different points on the spin sphere[62].

## Derivation of zero mode

The zero mode is obtained by analysing the symmetries of the Hamiltonian presented in the supplementary material of a previous report[32]. Without dipole–dipole interactions, the energy density of the bulk system in equation (1) is invariant under the following transformations from $\mathbf{m}'(\mathbf{r}')$ to $\mathbf{m}(\mathbf{r})$ and vice versa:

$$\mathbf{m}(\mathbf{r}) = \begin{pmatrix} \cos(kz_0) & -\sin(kz_0) & 0 \\ \sin(kz_0) & \cos(kz_0) & 0 \\ 0 & 0 & 1 \end{pmatrix} \cdot \mathbf{m}'(\mathbf{r}'), \tag{21}$$

where

$$\mathbf{r}' = \begin{pmatrix} \cos(kz_0) & \sin(kz_0) & 0 \\ -\sin(kz_0) & \cos(kz_0) & 0 \\ 0 & 0 & 1 \end{pmatrix} \cdot \mathbf{r} - z_0\hat{\mathbf{e}}_z, \tag{22}$$

and $z_0$ is an arbitrary parameter, which, in the most general case, defines the screw-like motion of an entire magnetic texture about the $z$ axis with pitch $2\pi/k = L_D$. Of note, there are at least two cases in which the transformation (equations (21) and (22)) does not affect the magnetic texture, meaning that $\mathbf{m}'(\mathbf{r}') = \mathbf{m}(\mathbf{r})$ holds for any value of $z_0$. The first case is rather trivial and corresponds to the conical phase with the wave vector aligned parallel to the $z$ axis (see equation (3)). The second case is particularly intriguing, and involves the skyrmion string in the conical phase, in which the primary axis of the string aligns with the rotation axis[32]. Applying the transformation in equations (21) and (22) to the skyrmion string with a hopfion ring describes the screw motion of the hopfion ring around the string representing a zero mode. The parameter $z_0$, in this case, denotes the displacement of the hopfion ring along the string. Supplementary Videos 7 and 8 provide a

visualization of such a screw motion of two different hopfion rings depicted in Fig. 4e,f. Evidence for such zero mode in other 3D solitons can be found in previous reports[8,63].

## Specimen preparation

FeGe TEM specimens were prepared from a single crystal of B20-type FeGe using a FIB workstation and a lift-out method[20].

## Magnetic imaging in the transmission electron microscope

The Fresnel defocus mode of Lorentz imaging and off-axis electron holography were performed in an FEI Titan 60-300 TEM operated at 300 kV. For both techniques, the microscope was operated in Lorentz mode with the sample at first in magnetic-field-free conditions. The conventional microscope objective lens was then used to apply out-of-plane magnetic fields to the sample of between −0.15 and +1.5 T. A liquid-nitrogen-cooled specimen holder (Gatan model 636) was used to vary the sample temperature between 95 and 380 K. Fresnel defocus Lorentz images and off-axis electron holograms were recorded using a 4k × 4k Gatan K2 IS direct electron counting detector. Lorentz images were recorded at a defocus distance of 400 μm, unless otherwise specified. Multiple off-axis electron holograms, each with a 4 s exposure time, were recorded to improve the signal-to-noise ratio and analysed using a standard fast Fourier transform algorithm in Holoworks software (Gatan). The magnetic induction map shown in Fig. 1 was obtained from the gradient of an experimental magnetic phase image.

Supplementary Videos 1–5 show in situ Lorentz TEM images captured at a defocus distance of approximately 400 μm and at a sample temperature of 180 K. Each video begins with several cycles of field swapping, in which the applied magnetic field alternates between positive and negative directions perpendicular to the plate. The field amplitude is limited to 50 mT or less. As the magnetic field in the transmission electron microscope is provided by the objective lens, this alternating field leads to a visible rotation of the image on the screen. Counter-clockwise rotation indicates an increase in the field towards the viewer and vice versa. These field-swapping cycles induce edge modulations that propagate towards the centre of the sample. The primary objective of this cycle is to generate edge modulations that propagate out of the free edges. After a few field-swapping cycles, closed loops near the centre of the square sample form. To enhance visibility, the playback speed of all of the videos has been doubled. Once at least one closed loop has been formed away from the sample edges, the magnetic field is increased to approximately 150 mT, resulting in the nucleation of a hopfion ring. Supplementary Video 1 illustrates the nucleation of a double hopfion ring.

Supplementary Video 4 reveals instabilities of the hopfion ring. After hopfion-ring nucleation, the magnetic field was initially decreased below a threshold value of approximately 50 mT, causing the hopfion ring to lose its shape and elongate over the sample. Subsequently, the field was increased again, leading to the reformation of a compact hopfion ring surrounding six skyrmion strings. Finally, the field was increased further above 190 mT, leading to the collapse of the hopfion ring. The concluding frame of Supplementary Video 4 depicts a cluster of six skyrmions without a hopfion ring.

## Data availability

Source data for TEM images are available at https://doi.org/10.5281/zenodo.8281078. Source data for micromagnetic simulations are provided with the paper.

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

**Acknowledgements** We thank H. Du for help with specimen preparation, and J. V. Vas and Q. Lan for their help with the video capturing of hopfion-ring nucleation. This project has received funding from the European Research Council under the European Union's Horizon 2020 Research and Innovation Programme (grant 856538; project '3D MAGiC'). F.Z. acknowledges financial support from the Fundamental Research Funds for the Central Universities and the National Natural Science Fund for Excellent Young Scientists Fund Program (Overseas); N.S.K. and S.B. acknowledge financial support from the Deutsche Forschungsgemeinschaft through SPP 2137 'Skyrmionics' (grants KI 2078/1-1 and BL 444/16-2, respectively); F.N.R. acknowledges support from the Swedish Research Council; and R.E.D.-B. is grateful for financial support from the European Research Council under the European Union's Horizon 2020 Research and Innovation Programme (grant 823717; project 'ESTEEM3') and the Deutsche Forschungsgemeinschaft (project ID 405553726; TRR 270).

**Author contributions** F.Z. and N.S.K. conceived the project and designed the experiments. F.Z., L.Y. and W.S. performed the TEM experiments and data analysis. N.S.K. and F.N.R. developed the theory and performed numerical simulations. F.N.R. performed homotopy-group analysis. N.S.K., F.Z. and F.N.R. prepared the manuscript. All authors discussed the results and contributed to the final manuscript.

**Funding** Open access funding provided by Forschungszentrum Jülich GmbH.

**Competing interests** The authors declare no competing interests.

**Additional information**
**Correspondence and requests for materials** should be addressed to Fengshan Zheng, Nikolai S. Kiselev or Filipp N. Rybakov.

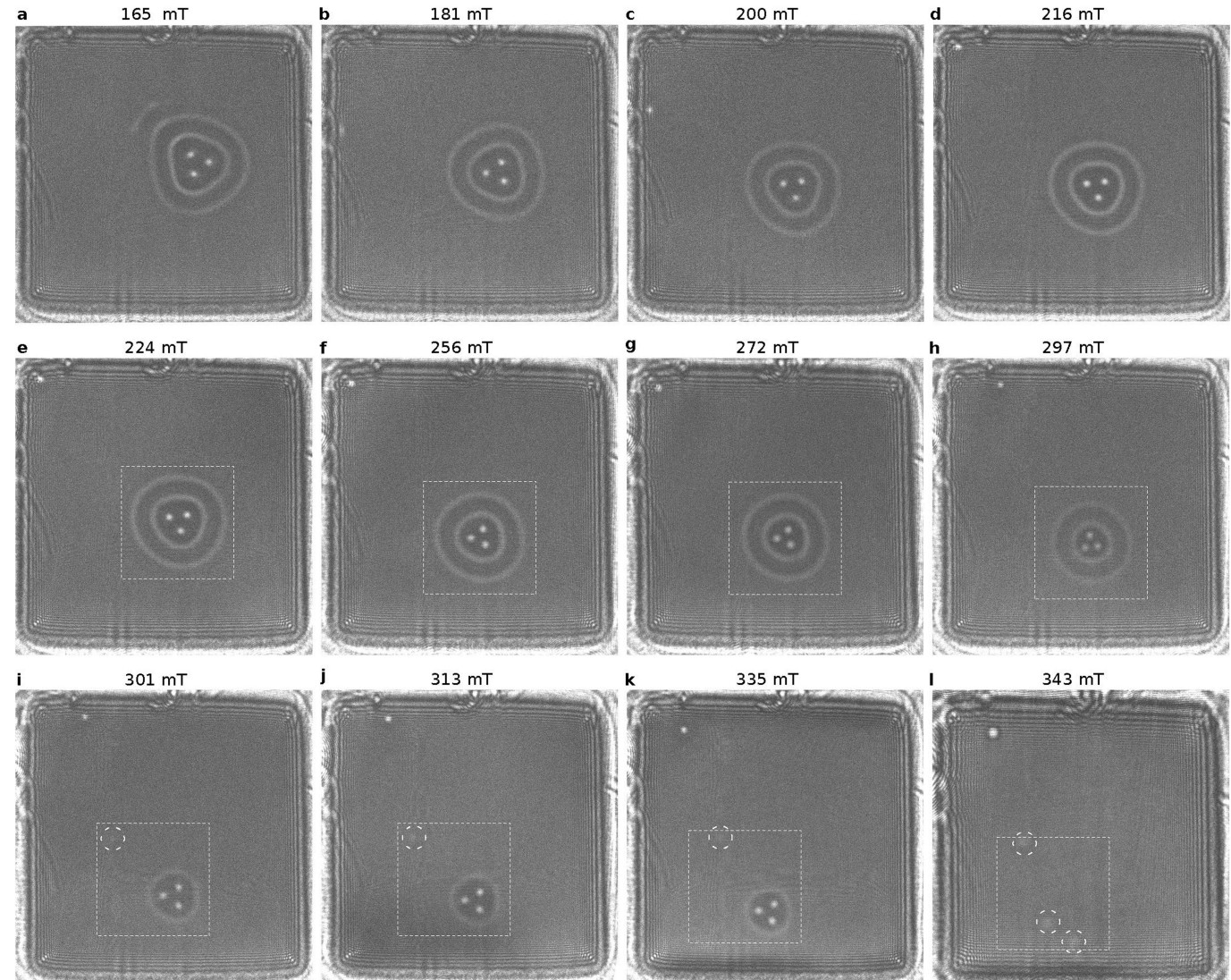

**Extended Data Fig. 1 | Field evolution of the magnetic state with double hopfion rings and three skyrmions.** This is an extended version of Fig. 2 (top row), showing over-focus Lorentz TEM images recorded at a specimen temperature of 95 K. In **a**, the skyrmion string is attached to the outer ring. In **b**, the skyrmion moves to the left edge of the sample but becomes more visible in **c** with increasing field. In **d**, the skyrmion moves to the top left-top corner. Between **e** and **f**, the texture rotates slightly. In **g** and **h**, one can observe compression of the rings. The outer hopfion ring collapses at 301 mT to either a chiral bobber or a dipole string, as marked by a dashed circle in **i**. In **j** and **k**, the single ring continues to contract with increasing field. The entire system collapses at 343 mT to three bobbers or dipole strings (see **l**).

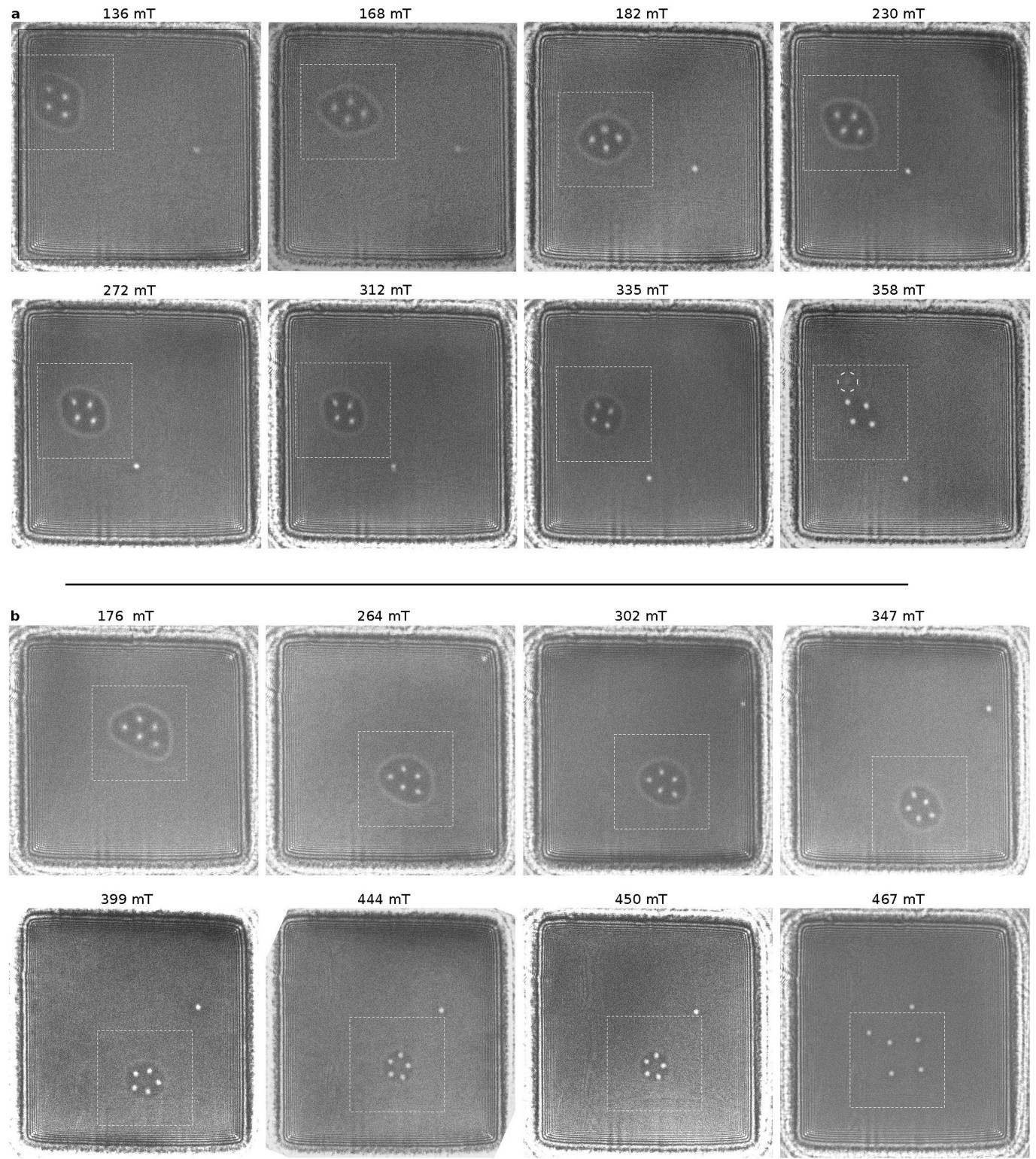

**Extended Data Fig. 2 | Field evolution of the magnetic state with a hopfion ring. a**, An extended version of Fig. 2 (second row), showing over-focus Lorentz TEM images of a hopfion ring around four skyrmions recorded at a specimen temperature of 180 K. The outer ring collapses at 358 mT to a chiral bobber or a dipole string, as marked by a dashed circle. **b**, An extended version of Fig. 2 (third row), showing over-focus Lorentz TEM images recorded at a specimen temperature of 180 K. At higher applied magnetic fields (399, 444 and 450 mT), skyrmions inside the hopfion ring have fivefold symmetry. The hopfion ring collapses at 467 mT, and the nucleation of a chiral bobber or a dipole string is not observed in this case.

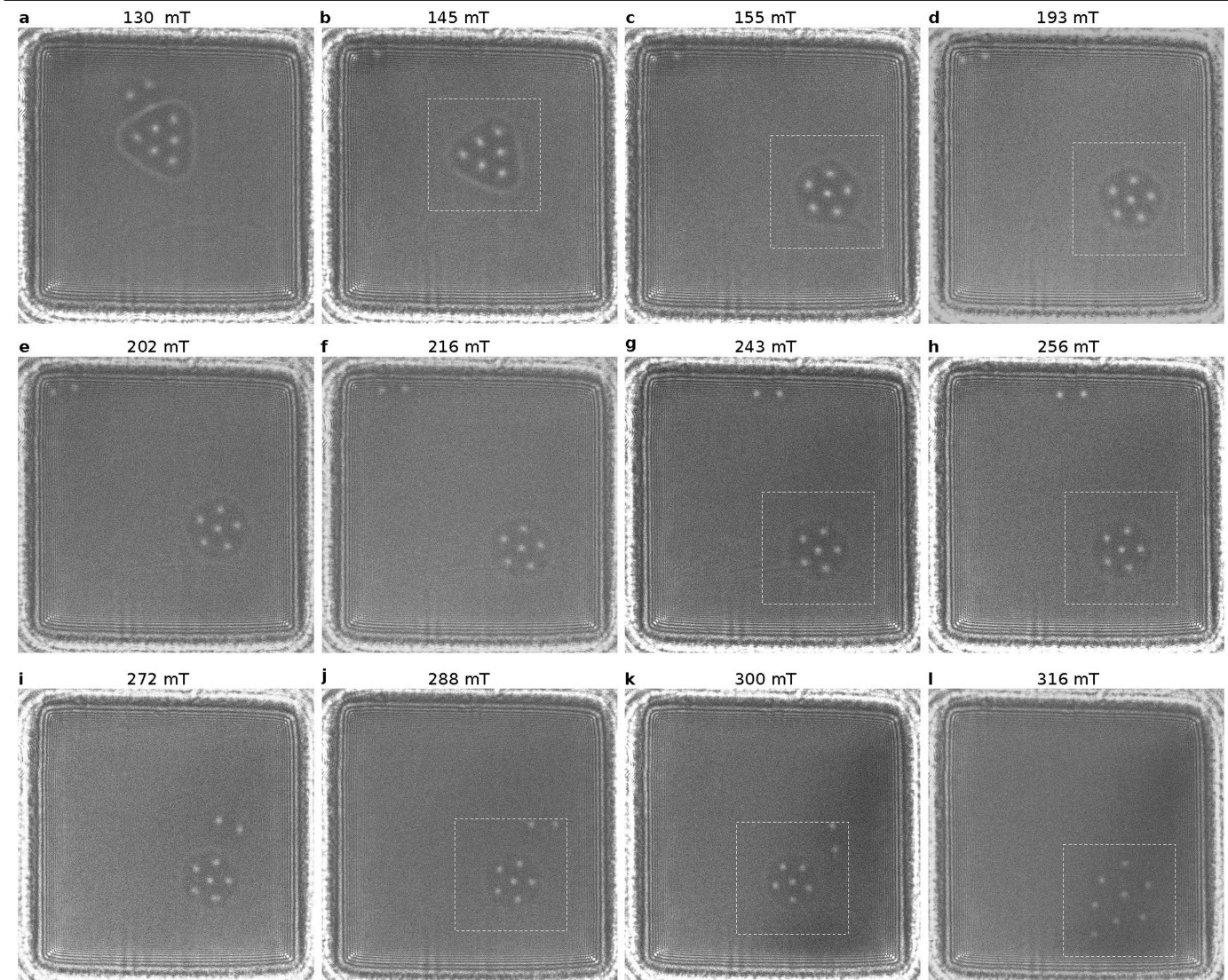

**Extended Data Fig. 3 | Field evolution of the magnetic state with a hopfion ring and six skyrmions.** This is an extended version of Fig. 2 (bottom row), showing over-focus Lorentz TEM images recorded at a specimen temperature of 200 K. In **a**, two skyrmions are attached to the hopfion ring. In **b**, skyrmions move to the left-top corner of the sample. In **c**, the skyrmion cluster surrounded by the hopfion ring takes a pentagon shape. With increasing field, in **d**, **e**, **f**, **g**, and **h**, the contrast of the hopfion ring gradually decreases. In **i**, **j**, and **k**, the skyrmions return to the sample center and interact with the hopfion ring. **l**, The hopfion ring collapses at 316 mT, and the nucleation of a chiral bobber or a dipole string is not observed in this case.

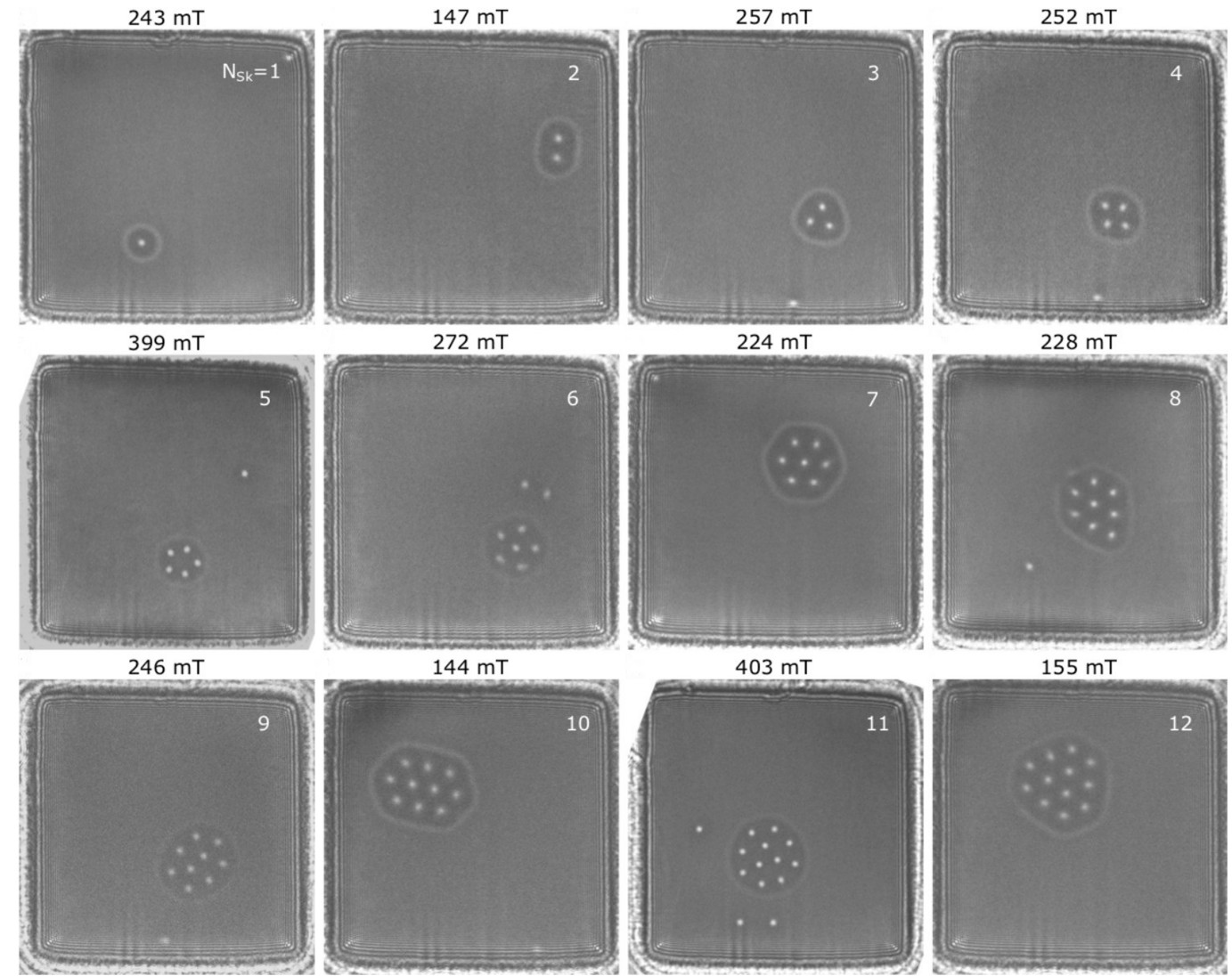

**Extended Data Fig. 4 | Lorentz TEM images of magnetic textures of topological charge −1 to −12 with a single hopfion ring.** Over-focus Lorentz TEM images are shown. The number of skyrmions inside the hopfion ring, $N_{Sk}$, is labelled in the upper right corner of each image. The applied magnetic field is indicated above each image. The images were recorded at a specimen temperature of between -180 and -200 K.

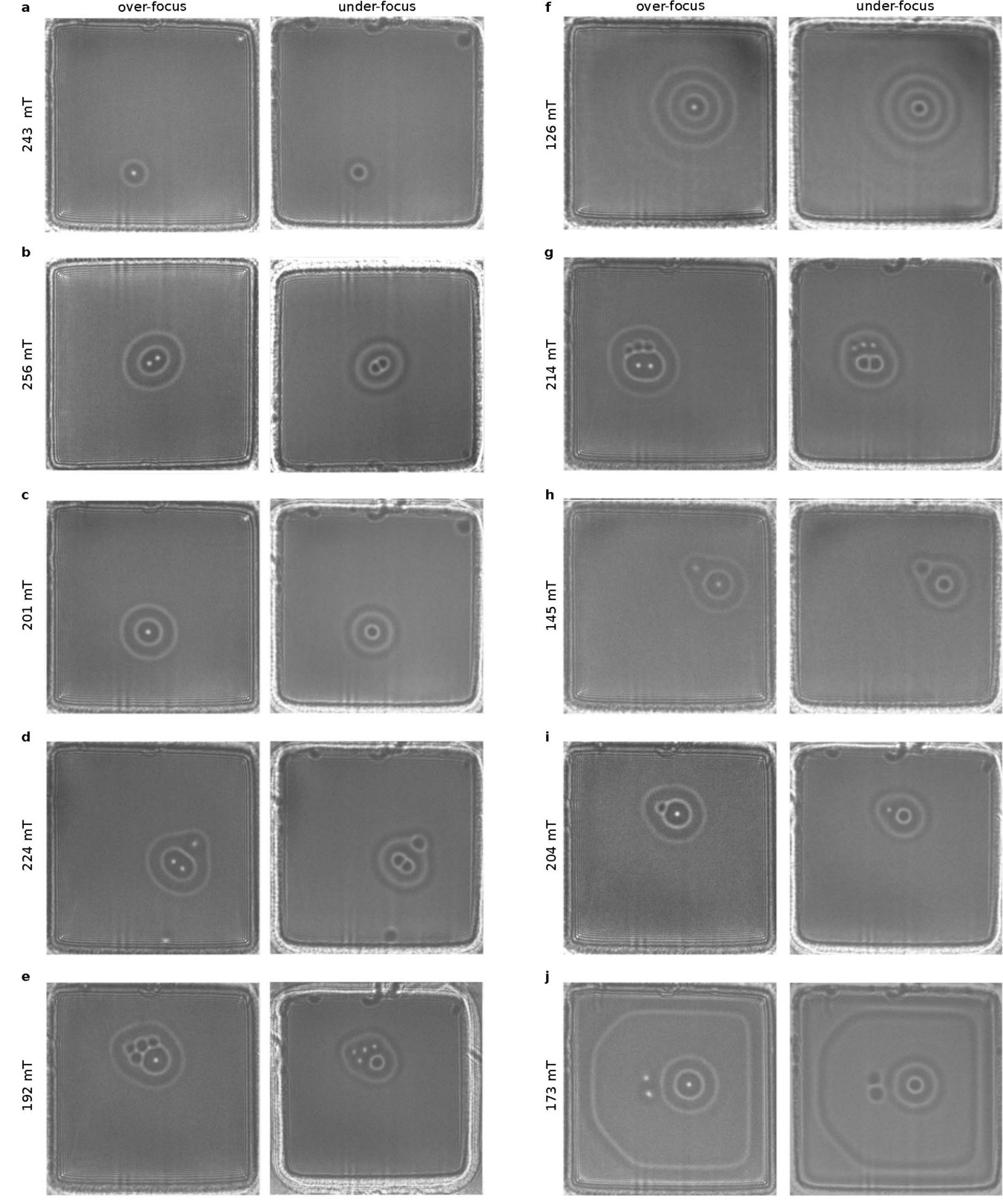

**Extended Data Fig. 5 | Examples of exotic magnetic states with hopfion rings and skyrmion strings.** Over-focus and under-focus Lorentz TEM images recorded at a specimen temperature of between −180 and −200 K are shown. The magnitude of the applied magnetic field is marked on the left of each image. The images in **a**, **b**, **d**, **e**, **g**, and **i** show the magnetic states depicted in Fig. 4 but for the entire field of view of the sample. **c** and **f** are the magnetic states with a single skyrmion string and multiple hopfion rings. The magnetic state in **h** is similar to that in **c** but with an extra skyrmion in-between internal and external hopfion rings. The intermediate state in **j** is similar to that shown in Fig. 1**f** at 190 mT.

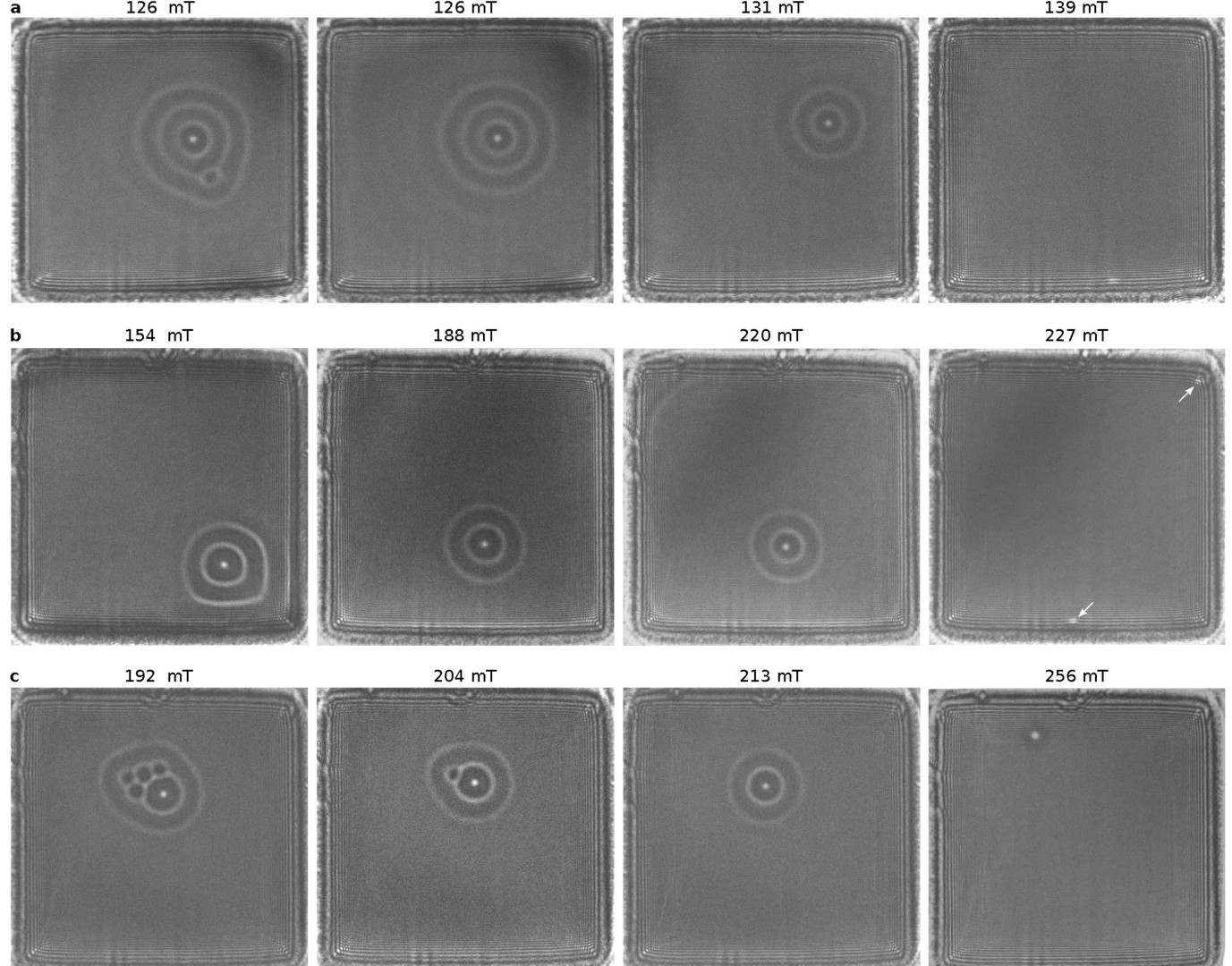

**Extended Data Fig. 6 | Field evolution of magnetic states with multiple hopfion rings.** The images are arranged in rows. **a**, Magnetic textures with three hopfion rings. The outermost hopfion ring collapses at 131 mT. The two inner hopfion rings collapse at 139 mT. **b**, Double hopfion rings coupled to one skyrmion string. The double rings show non-homogeneous contrast at lower applied fields owing to interaction with the specimen edge. With increasing applied field, the ring detaches from the specimen edge and the two hofion

rings show homogeneous contrast. The two hopfion rings collapse at 227 mT. **c**, The magnetic state of a positive topological charge surrounded by a single hopfion ring. With increasing field, the state evolves into a skyrmion surrounded by two hopfion rings at 213 mT and collapses to a single skyrmion at 256 mT. The applied magnetic field is labelled above each image. The Lorentz TEM images were recorded over-focus at a defocus distance of 400 μm at a specimen temperature of 95 K.

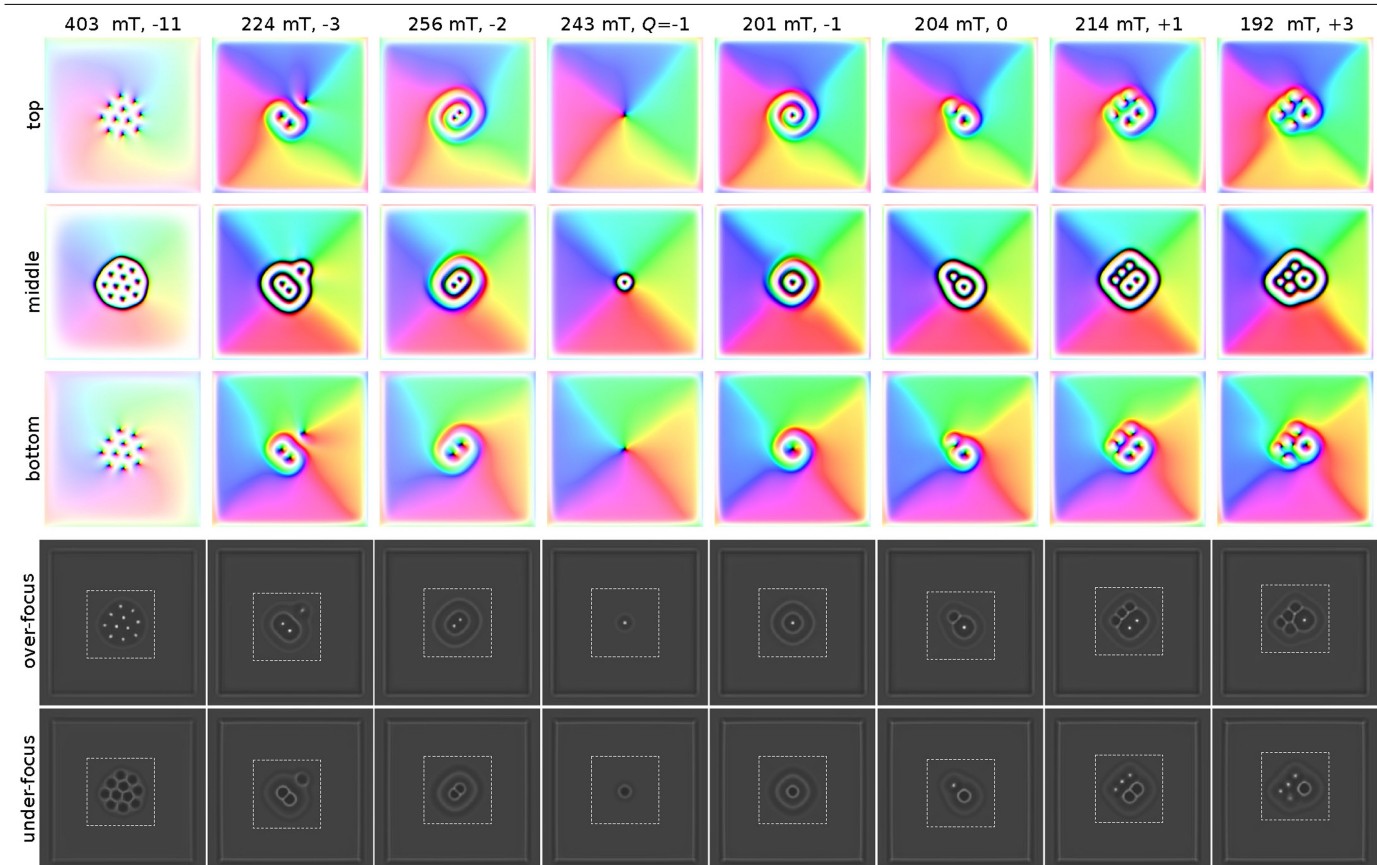

**Extended Data Fig. 7 | Micromagnetic simulations of exotic magnetic states with hopfion rings.** Each column represents a single hopfion ring obtained by energy minimization, assuming open boundary conditions. The calculations were performed for the applied magnetic fields indicated above each column. The first three rows show the magnetization vector field in the top, middle and bottom planes of the square sample, whose size (1 μm × 1 μm × 180 nm) is identical to that used in the experiment. An alternative representation of the magnetization vector fields of the magnetic states by means of isosurfaces is provided in Fig. 4 in the main text (for the magnetic states shown in the first four columns) and Extended Data Fig. 8 (for the magnetic states shown in columns 5 to 8). The two bottom rows show theoretical over-focus and under-focus Lorentz TEM images calculated for a defocus distance of 400 μm. The dashed squares indicate the field of view that was used in Fig. 3 in the main text.

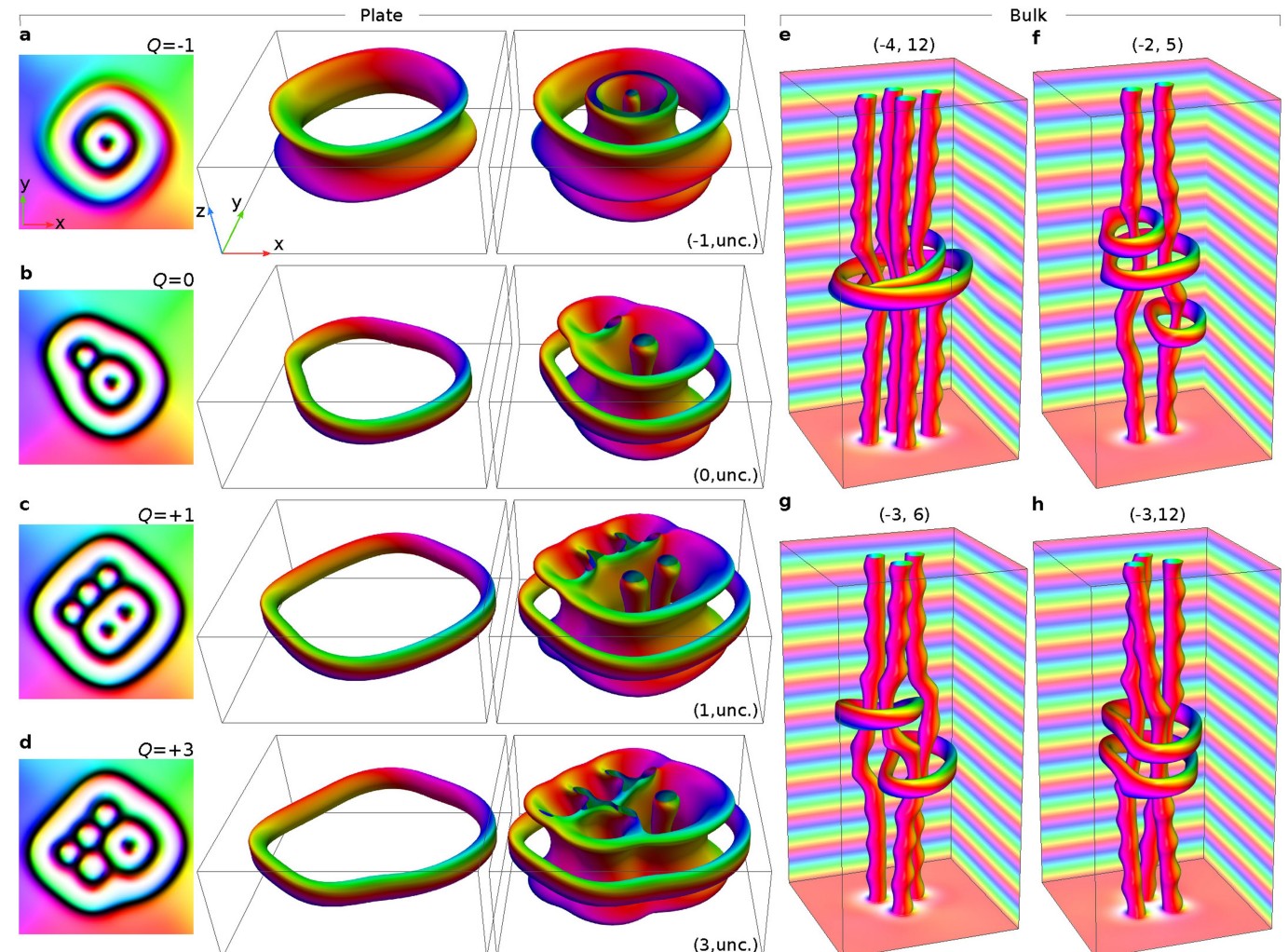

**Extended Data Fig. 8 | Micromagnetic simulations of magnetic textures with hopfion rings in a confined sample and in the bulk. a–d**, As in Fig. 4a–d, but for the other experimentally observed magnetic states shown in Fig. 3e–h with $Q = -1$, 0, +1 and +3. **e–h**, Different configurations of coupled skyrmion strings and hopfion rings in the bulk. The skyrmion–hopfion index given by the ordered pair of integers $(Q, H)$ is specified for each state. The hopfion rings in **e**

are interlinked. In **f**, two hopfion rings (top and bottom) are located on individual strings, and one hopfion ring (in the middle) surrounds two strings. In **g**, two hopfion rings share one skyrmion string and have one extra individual string. The braiding of the skyrmion strings in **h** increases the hopfion index compared to the state shown in Fig. 4**h**. Note that, for the states shown in **a–d**, the Hopf charge is uncertain (see the main text for details).

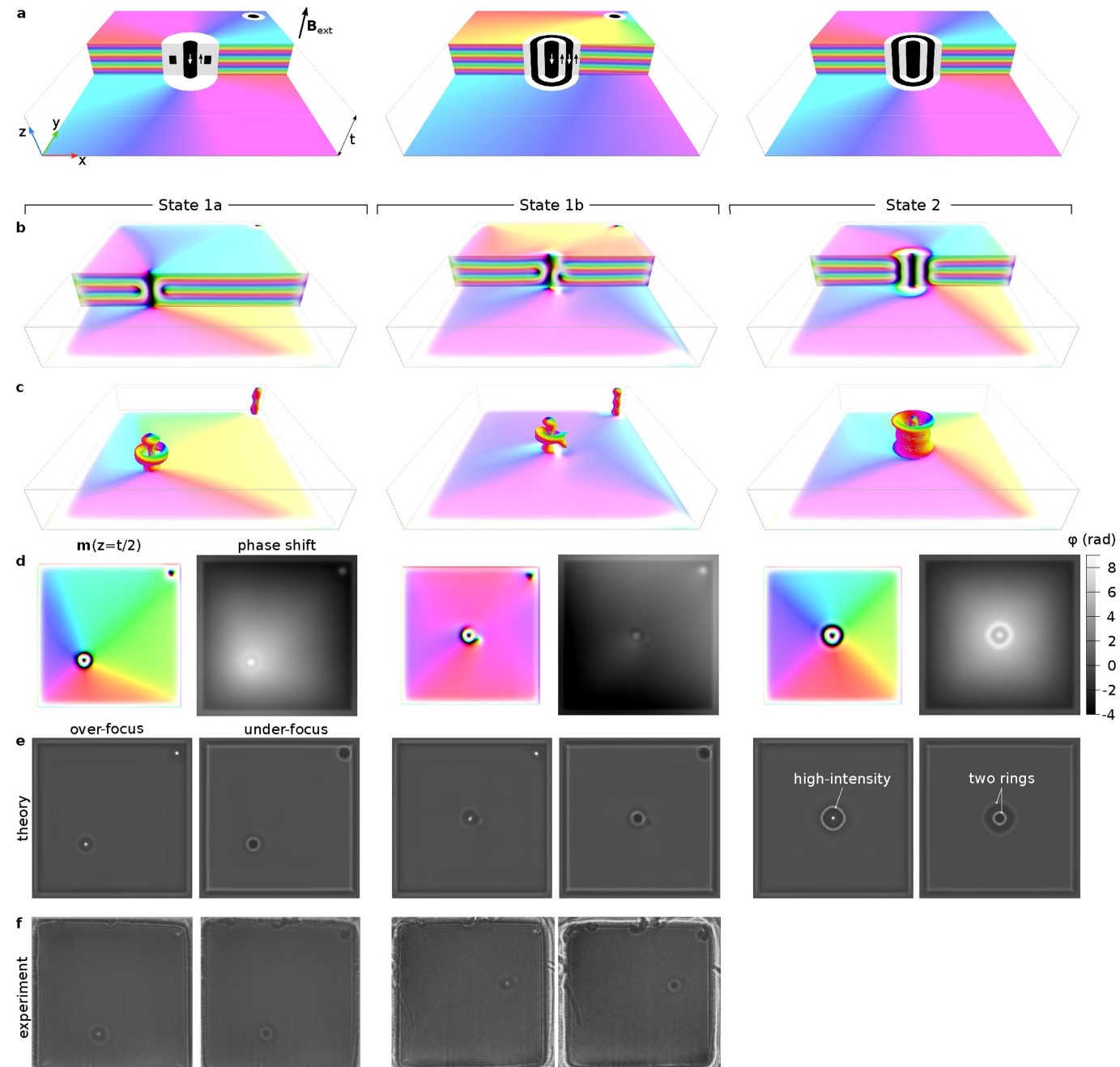

**Extended Data Fig. 9 | Micromagnetic simulations for hopfion rings, with modulations penetrating the whole thickness, and a comparison of the contrast with experimental measurements. a**, Three initial states. **b,c**, Corresponding magnetic textures after energy minimization. In **b**, a cross-cut of the simulated state is shown; in **c**, the same magnetic texture is visualized using isosurfaces $m_z = 0$. The calculations for all three cases were conducted for an external magnetic field of 243 mT, with a very small tilt angle not exceeding 2°. For details, see Mumax Script II in the Supplementary information. **d**, Corresponding cross-cuts for the median plane and theoretical phase images calculated for each of the three states. **e**, Corresponding theoretical Lorentz TEM images, assuming a defocus distance of 400 μm, as in the experimental

set-up. **f**, Experimental Lorentz TEM images recorded in an external magnetic field of 243 mT and at a defocus distance of 400 μm. The experimental images match the theoretical images well for States 1a and 1b with a compact hopfion ring. In State 1a, the vortex is centred on the hopfion ring, whereas in State 1b this is not the case. Note the bumps on the hopfion ring in State 1b visible in the experimental and theoretical Lorentz images. For State 2, which exhibits magnetic modulations extending along the thickness, the theoretical Lorentz TEM images show two distinct features: a strongly contrasting bright ring in the over-focus image and an additional bright ring in the under-focus image. No high-intensity bright ring in the over-focus image or second bright ring in the under-focus image are observed in the experimental images.

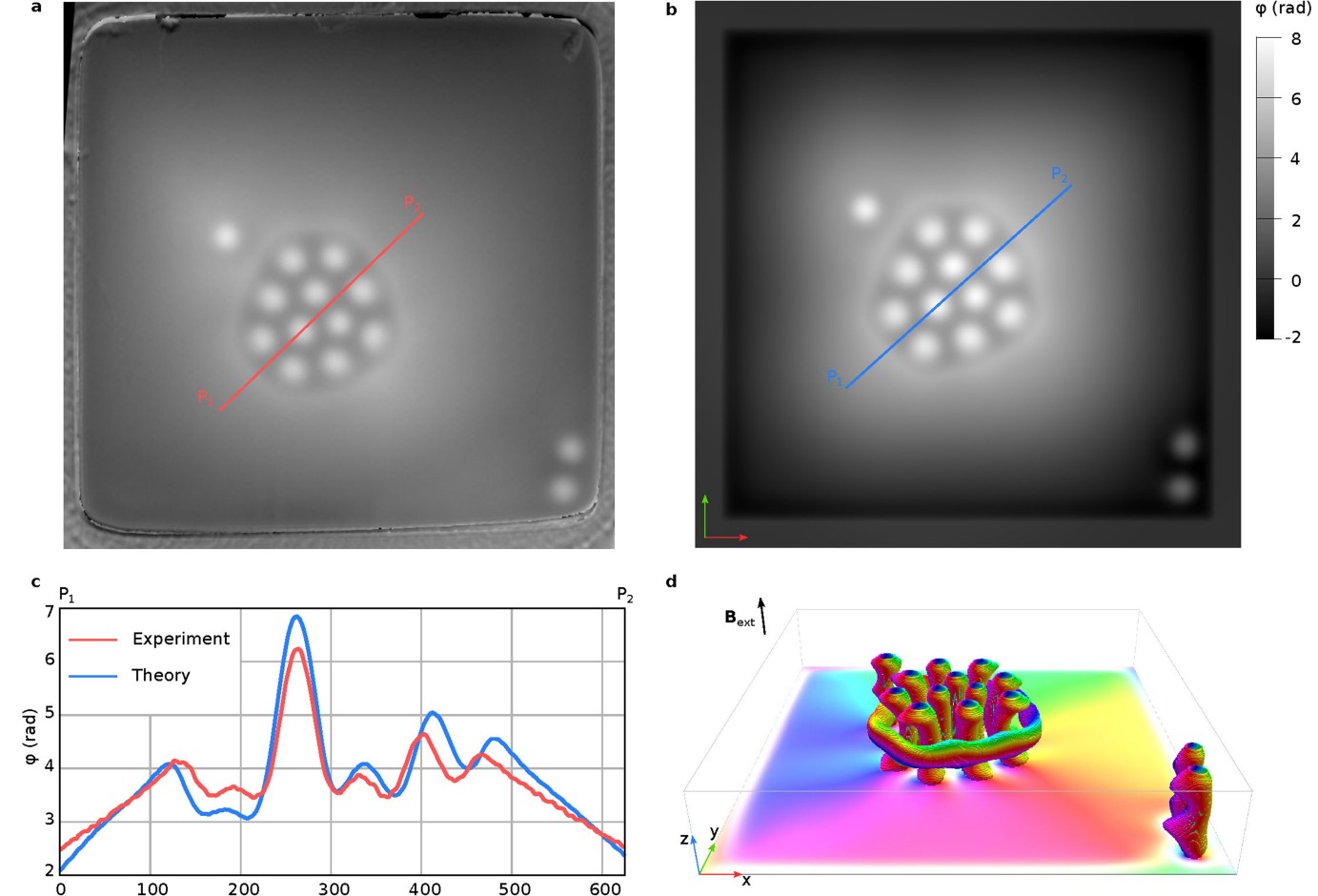

**Extended Data Fig. 10 | Quantitative comparison of experimental and theoretical phase-shift images. a**, Experimental electron optical phase-shift image recorded using off-axis electron holography at $T = 180$ K in an external magnetic field of 200 mT. **b**, Theoretical phase-shift image calculated for the equilibrium configuration in the micromagnetic model, as shown in **d**. The grey levels used to display the magnitude of the phase shift, $\varphi$, are identical for the experimental and theoretical images. **c**, Line profiles of the phase shift extracted along the paths marked in **a** and **b**. The experimental and theoretical profiles indicate that the skyrmion inside the hopfion ring induces an electron optical phase shift of approximately 3.5 radians, whereas the hopfion ring induces an electron optical phase shift of approximately 1.0 radians. **d**, Magnetic texture obtained from micromagnetic simulations for an external magnetic field of 200 mT. The magnetization field is represented by the isosurface $m_z = 0$.