## [Peer Review File · Nature]

Manuscript Title: Hopfion rings in a cubic chiral magnet

Reviewer Comments & Author Rebuttals

Reviewer Reports on the Initial Version:

Referees' comments:

Referee #1 (Remarks to the Author):

In this paper, the authors present their work on creating and stabilizing complex magnetic spin textures comprising of hopfion rings and skyrmion strings in patterned thin plates of FeGe. They identify a specific magnetic field protocol to stabilize these spin textures based on micromagnetic simulations. Then they provide experimental evidence based on Lorentz TEM images and electron holography data that maps the projected magnetic induction maps. Overall there is good agreement between the micromagnetic simulations and experimental data presented. However, the novelty of this work does not match the standard required by Nature journal. There are several ambiguities in the results that need to be clarified. Moreover, only the existence of these states has been shown, but apart from that there is a lack of understanding that is presented in the results about reasons for their stability, and ability to control them. Therefore, publication of this work is premature at this stage.

Below are several comments and questions about this work.

- 1) The authors have presented the simulated data on how field swapping protocol nucleates the novel spin textures. However, experimental data that follows this field swapping protocol should also be shown to clearly identify the switching of the magnetization directions.
- 2) A major concern is the contrast arising in the Lorentz TEM images from the hopfion rings. Since this is a projection method of imaging, the net result is due to integrated magnetic induction throughout the thickness of the sample. Therefore, the contrast observed is very similar to a Bloch domain wall that extends in 3D. The authors should consider clarifying this ambiguity, to conclude that the observed contrast is solely due to hopfion rings.
- 3) Although the authors do refer that their observations are different than the skyrmion bags that have been observed before, but overall the difference appears to be minimal. They have presented some homotopy analysis but that is restricted to skyrmion topologies. How does that inform us about the hopfion rings? What is the fundamental difference between the two structures (rings and bags) is not clear. Since the twisted structure they are referring to only appear in the central plane of the 3D plate, how this is different that the ones referred to as skyrmion bags is also not apparent.
- 4) To this point, a secondary signature of the hopfion ring should be presented to confirm their twisted spin texture either 3D mapping of magnetic field (or atleast images at different tilts to show the effect of twist and differentiate from simple Bloch wall texture) or electrical driven signature.

5) In figure 1, the color representation is quite confusing because it does not represent solely a single magnetic vector (m_x , m_y or m_z) but changes between various subpanels. For example, in the 3D renderings it is not clear what does the color represent?

6) For figure 2, although the external field is given as a grayscale bar on top, numerical values would be more useful. Also there is some missing text on the bottom most row for $Q=6$.

7) For figure 6, the last column with different opacity is also not terribly useful. What was the purpose of showing this?

8) The formatting of some references is not correct. For e.g. reference 32 is missing the title.

Referee #2 (Remarks to the Author):

The authors Zheng et al. have submitted a highly interesting manuscript in which they analyze the stability of magnetic hopfion rings. The topic is relevant and the experimental and theoretical data is of excellent quality. Therefore, the manuscript should be published in a high-impact journal.

However, currently, I doubt that the study deserves publication in Nature because of a lack of significance. My main concern is the following:

1. This is not the first observation of hopfions. As the authors correctly write in the introduction, magnetic hopfions have already been observed in Ref. 24. The authors distinguish the hopfions in confined geometries, as presented in that reference, from their observation of hopfions that exist as rings around one or multiple skyrmion strings. The authors label them 'hopfion rings'. I understand that the stabilizing mechanism is different but why is that of crucial importance? Fundamentally, I do not see a higher significance of hopfion rings, especially since they can only exist around skyrmion strings. In terms of applications, I do not see that great of an advantage either. For example, in terms of current-driven motion, hopfions have been predicted to not suffer from the skyrmion Hall effect. However, in combination with the skyrmion string(s) the skyrmion Hall effect should be present.

I am looking forward to read the authors' reply. Please elaborate on the drastic improvement of hopfion strings over conventional hopfions that justifies a publication in Nature. After all, Ref. 24 was published in Nature Communications, so why should this study be published in Nature?

Further minor comments are as follows:

2. The authors mention a damaged layer in the experiment as well as the simulation. Is it possible to stabilize the hopfions without that damaged layer? How can we understand the importance of the damaged layer? Can you somehow 'quantify' the damaged layer so that other groups will be able to reproduce the results in experiments?

3. The authors strongly distinguish their hopfion rings from the hopfions observed in confined geometries, as discussed in issue 1. However, later in the paper they write that the confined geometry is important for the generation of the hopfion rings (but not for the stability). I am not sure if I understand the authors correctly but, in my understanding, this should eliminate all advantages of hopfion strings versus the hopfions in confined geometries because without a confined geometry you are not able to nucleate the hopfion rings in the first place.

4. If a hopfion ring forms around two or more skyrmion strings, the strings are braided. Is the braiding essential for the stabilizing mechanism or could the hopfion ring form around straight skyrmion strings as well?

5. In Fig. 3 a hopfion ring around a single skyrmion string is labeled an 'exotic state'. From a naïve approach this state should be the simplest version of a hopfion ring. Why is it 'exotic'? Is the probability for the nucleation of this type of hopfion ring smaller than for other types?

6. About the probabilistic character of the nucleation: I think the study would benefit from quantifying the probability to nucleate a hopfion ring. What are the chances that a hopfion ring is nucleated per alternation of the magnetic field direction? Are there ways to increase this percentage? How does it depend on the number of skyrmions inside of the hopfion? How does it depend on temperature?

7. Concerning the last point: The authors write that the protocol becomes more reliable at higher temperatures and that it requires more field swapping cycles to become stable at lower temperatures. How can we understand this?

8. Please clarify whether or not a hopfion ring always remains stable if the magnetic field orientation is continued to be alternated after a hopfion ring has successfully been nucleated. This would help to assure that a hopfion ring forms with a high probability in potential devices because you could simply change a field direction many times.

9. The authors write that Fig. 2 shows that the symmetry changes reversibly. What exactly does that mean? The process in the top row is certainly not reversible.

10. Out of interest: Do you think it is possible to better analyze these objects by three-dimensional techniques such as holography? If not, what are the problems?

Referee #3 (Remarks to the Author):

In their paper „Hopfion rings in a cuboc chiral magnet“ the authors Fengshan Zheng et al. present evidence for the existence of so-called Hopfions using Lorentz Transmission Electron Microscopy (LTEM) and electron holography compared to micromagnetic simulations. Hopfions are three dimensional solitons predicted to be present in three dimensional chiral magnets. In contrast to skyrmion strings which (ideally) penetrate the whole specimen, Hopfions are truly localized in space,

detached from sample boundaries and thus are discussed as topologically protected information carriers. The authors provide a special magnetic field protocol applied to thin FeGe platelets that allows controlled creation of stable magnetic structures with different topological comprising of Hopfions (essentially a closed string) wrapped like a belt around skyrmion tubes with various topological numbers. While Hopfions in magnetic thin film materials might have been detected previously in ref 24 (even though in this paper the evidence is not very strong) this is the first observation of what the authors call Hopfion rings in cubic chiral magnets. This is a beautiful paper and I favour publication in Nature.

I would like to ask the authors to comment on the following minor issues:

In the abstract potential applications are mentioned, but the authors do not expand on this. I suggest removing this half sentence or expanding the discussion concerning applications in the main text.

In the caption of Fig. 1 please correct the dimensions of the platelet and please add information concerning the sample temperature.

Please provide more details concerning the sample preparation using FIB and the damage layer and move this part to the main text, this seems to be important information.

Chiral bobbles are mentioned e.g. in the caption of Fig. 2. Could the authors comment on their stability? Do they occur at defects, or why are they stable?

On page 3 right side bottom. The authors mention that intermediate configurations are observed seldomly. What does seldom mean? How many instances have been observed in how many experiments in total?

At the bottom of page three the authors mention that thermal fluctuations have an influence on the nucleation induced by the magnetic field protocol. Can this be corroborated in temperature dependent simulations (see also Fig.3 for simulations of images taken at different temperatures)?

Did the authors use different platelet sizes (in-plane size)? Does this have an influence on nucleation?

Can the authors move the composite particles with tilted magnetic field? The authors mention that tilt angles should not exceed 5 degrees for nucleation (this information might also be moved to the main text if space allows), but what happens if the particle has been nucleated and the field is tilted?

In Fig. S1a a section of a ring is visible with slightly higher contrast at the ends, why is this so? Can rings rip? Or is this section pinned by defects?

Author Rebuttals to Initial Comments:

Point-by-point response to the reports from the reviewers:

We are very grateful to all three reviewers for their constructive comments. We believe that we have addressed all of their concerns and questions, in particular about the novelty of our work. We have conducted new experiments and carried out additional numerical simulations to strengthen both the experimental and the theoretical parts of our manuscript.

For the convenience of the editor and reviewers, the major changes that we have made in the revised manuscript are as follows:

1. A new co-author (Dr. Wen Shi, Forschungszentrum Jülich) has been added, due to her contribution to new experiments during the revision process. All of the previously listed co-authors have agreed to the change in co-authorship.
2. Eight supplementary videos have been added to illustrate the process of hopfion ring creation in our experiments (Videos 1-5) and to illustrate the intriguing properties of hopfion rings (Videos 6-8).
3. In-depth analysis and discussion have been added about the topology and homotopy of the solutions discovered in our work. A rigorous homotopy classification, which accounts for the hopfion rings and corresponding skyrmion-hopfion topological charge, has been introduced.
4. The content of Fig. 4 in the main text and Extended Data Fig. 10 has been modified. Following the suggestion of Reviewer #1, the redundant panel showing isosurfaces of magnetic textures with opacity has been deleted and results for hopfion rings calculated for bulk systems have been added. Extended Data Fig. 8 has been redesigned to make it more readable.
5. Improvements have been made to the text of the manuscript. Redundancy and typos have been corrected. The new and revised parts of the text are highlighted in red in a separate document.

A point-by-point response to the reviewers' questions and comments is provided below.

Reviewer #1 (Remarks to the Author):

In this paper, the authors present their work on creating and stabilizing complex magnetic spin textures comprising of hopfion rings and skyrmion strings in patterned thin plates of FeGe. They identify a specific magnetic field protocol to stabilize these spin textures based on micromagnetic simulations. Then they provide experimental evidence based on Lorentz TEM images and electron holography data that maps the projected magnetic induction maps. Overall there is good agreement between the micromagnetic

simulations and experimental data presented. However, the novelty of this work does not match the standard required by Nature journal. There are several ambiguities in the results that need to be clarified. Moreover, only the existence of these states has been shown, but apart from that there is a lack of understanding that is presented in the results about reasons for their stability, and ability to control them. Therefore, publication of this work is premature at this stage.

Response: We sincerely appreciate the referee's positive comments about our experimental results and about their agreement with our micromagnetic simulations.

In response to the referee's remarks about novelty, we would like to emphasize that our work represents an essential advance in the field of complex magnetic spin textures. We have discovered a magnetic hopfion that has never been observed before in magnetic crystals. Although theoretical investigations about hopfions have been ongoing for several decades, their experimental realization in crystals has remained elusive. The fact that world-leading teams actively continue to search for hopfions is a testament to the significance of the problem.

In order to illustrate the complexity and competitiveness of the field, we would like to refer to recent works by Donnelly et al. (Nature Physics, 2021) and Kent et al. (Nature Communications, 2021). At the 18th minute of Donnelly's presentation available at <https://www.youtube.com/watch?v=QNcl9AXwl9A>, the authors express their disappointment about not finding hopfions. However, in their paper they acknowledge their potential existence. Kent et al. employed reverse engineering to create a synthetic material that artificially favored the stability of a single hopfion. However, it is important to note that such an approach is different from an observation in a natural crystal system. In contrast, our work, which focuses on the discovery and stabilization of hopfion rings in a magnetic crystal, represents a truly unique contribution to the field and a turning point in the field of magnetic hopfions.

In the revised version of the manuscript, we cite the papers of both Donnelly et al. (Nature Physics, 2021) and Kent et al. (Nature Communications, 2021), which provide valuable insight into the exploration of 3D magnetic textures, in the abstract.

With regard to stability and control of hopfions, we would like to emphasize that our hopfion rings never appear as the ground state of the system. Therefore, additional efforts are required to create and stabilize them. Just as for other magnetic solitons, a competition between different energy terms in the Hamiltonian plays a crucial role in the stabilization of hopfion rings. We have a thorough understanding of these stability mechanisms, and have provided detailed discussions on this topic in the revised manuscript.

In response to the referee's concerns about ambiguities in our results, we have made significant revisions to the manuscript to provide a more transparent and comprehensive picture of the hopfion rings and their properties. We have included a detailed discussion about the topological aspects of our discovery, and have provided further experimental information and accompanying videos, which illustrate the process of hopfion ring creation and exclude any ambiguities and provide a solid foundation for our findings.

We believe that our responses and revisions to the manuscript address all of the concerns raised by the reviewer. We are confident about the novelty, significance and clarity of our work.

Below are several comments and questions about this work.

1) The authors have presented the simulated data on how field swapping protocol nucleates the novel spin textures. However, experimental data that follows this field swapping protocol should also be shown to clearly identify the switching of the magnetization directions.

Response: In the revised version of the manuscript, we have provided five supplementary videos, which illustrate the process of hopfion ring creation. We have also added the following paragraph to the main text of the manuscript:

“The process of hopfion ring nucleation is demonstrated in Supplementary Videos 1-5. These videos were captured *in situ* at a temperature of $T = 180$ K. We performed several field swapping cycles in the initial stage with a small amplitude of approximately ± 50 mT. This step was designed to generate edge modulations that formed closed loops and propagated towards the center of the sample. Once one or a few of these loops had been created, we gradually increased the applied magnetic field up to approximately 150 mT, resulting in the formation of various hopfion rings.”

Moreover, we have added detailed descriptions of the videos in the Methods section.

2) A major concern is the contrast arising in the Lorentz TEM images from the hopfion rings. Since this is a projection method of imaging, the net result is due to integrated magnetic induction throughout the thickness of the sample. Therefore, the contrast observed is very similar to a Bloch domain wall that extends in 3D. The authors should consider clarifying this ambiguity, to conclude that the observed contrast is solely due to hopfion rings.

Response: We fully understand this concern and thought that this aspect had already been addressed in the original manuscript. In Extended Data Fig. 8, we compared the contrast of a compact hopfion ring and modulations extended over the thickness to our experimental data. We have updated Extended Data Fig. 8 to further clarify this point. The revised version of this figure clearly explains the difference between hopfion rings and modulations that extend over the thickness. It is important to emphasize that our micromagnetic simulations were performed without any parameter fitting, using micromagnetic constants for FeGe from Zheng et al. (Nature Nanotechnology, 2018). All other parameters, such as field, thickness and defocus distance, matched the experimental conditions.

We have also provided several supplementary videos that illustrate the difference in contrast between skyrmion bags surrounded by helical modulations and hopfion rings localized in a smaller sample volume. The two images shown below are snapshots taken from Supplementary Video 1.

The image on the left illustrates a skyrmion bag composed of four skyrmions surrounded by 360° Bloch domain walls that extend across the thickness of the sample. The image on the right shows double hopfion rings, which have much weaker contrast.

3) Although the authors do refer that their observations are different than the skyrmion bags that have been observed before, but overall the difference appears to be minimal. They have presented some homotopy analysis but that is restricted to skyrmion topologies. How does that inform us about the hopfion rings? What is the fundamental difference between the two structures (rings and bags) is not clear. Since the twisted structure they are referring to only appear in the central plane of the 3D plate, how this is different that the ones referred to as skyrmion bags is also not apparent.

Response: In the revised version of the manuscript, we have extensively examined the distinctive features of hopfion rings, highlighting their fundamental differences from skyrmions and skyrmion bags.

The sketch shown below explains this difference schematically.

First, we have expanded the section on homotopy analysis in the Methods section, providing a more comprehensive discussion about the homotopy group. We have introduced the skyrmion-hopfion topological charge and included it in the revised

version of Fig. 4 and Extended Data Fig. 10, and added corresponding discussions in the main text.

Second, it is important to note that skyrmion bags are primarily two-dimensional structures extending through the plate's thickness. In contrast, hopfion rings are fully localized in three dimensions, introducing new and intriguing physics that sets them apart from skyrmion bags. We have illustrated this distinction by presenting a wide range of solutions for bulk systems, and by providing rigorous theoretical results about the zero mode for hopfion rings, corresponding to the rotational-translational movement of the hopfion along the string. Supplementary Videos 7 and 8 illustrate such motion, and details are provided in the Methods section.

4) To this point, a secondary signature of the hopfion ring should be presented to confirm their twisted spin texture either 3D mapping of magnetic field (or atleast images at different tilts to show the effect of twist and differentiate from simple Bloch wall texture) or electrical driven signature.

Response: The reconstruction of a 3D magnetic texture typically involves conducting experiments in zero external magnetic field. In contrast, in the case of a hopfion ring or skyrmion it is essential to maintain a fixed direction of the external magnetic field relative to the sample.

Numerous efforts have been made to develop experimental techniques that can be used to measure and visualize 3D spin textures, typically by recording and analyzing tilt series of phase contrast images, diffraction patterns or spectra recorded using electrons, X-rays or neutrons. In transmission electron microscopy, since the conventional objective lens is currently used to apply a magnetic field to the specimen, it is not possible to tilt the field together with the specimen. A change in the direction of the field relative to the sample causes a change in the magnetic texture. Although it is in principle possible to use a magnetizing holder to tilt the applied field and the specimen together, this approach has not been developed reliably so far. It is therefore not yet technically possible to meet this requirement experimentally.

Nevertheless, we would like to emphasize that our results provide remarkable agreement between experimental observations and micromagnetic simulations, leaving no room for alternative explanations about the weak contrast of the rings that we observe. In addition to the rings themselves, we have successfully replicated the process of hopfion ring nucleation using micromagnetic modeling. By using micromagnetic modeling, we have reproduced more than twenty distinct textures, demonstrating high quantitative correspondence with our experimental observations.

In the revised version of the manuscript, we have included Supplementary Videos from new experiments, which provide additional evidence to support our findings.

We would like to note that Reviewer #2 raised a similar question, but labeled it "*out of interest.*" Reviewer #3 did not request additional experiments to provide additional confirmation of the hopfion ring.

5) In figure 1, the color representation is quite confusing because it does not represent solely a single magnetic vector (m_x , m_y or m_z) but changes between various subpanels. For example, in the 3D renderings it is not clear what does the color represent?

Response: In our manuscript, we use a unified color code to represent directions in three-dimensional space based on the HSV (hue, saturation, value) color space. This color coding scheme is utilized widely in various domains and has been adopted by numerous authors, including those whose works we reference. We acknowledge that the previous caption accompanying Fig. 1 did not explain this color code adequately. We have therefore revised the caption to Fig. 1, in order to provide a more comprehensive description of the color coding scheme.

6) For figure 2, although the external field is given as a grayscale bar on top, numerical values would be more useful. Also there is some missing text on the bottom most row for $Q=6$.

Response: Since the images shown in Fig. 2 were recorded at different temperatures, the exact values of the field may look confusing to readers. The only option to avoid this confusion would be to indicate the temperature in each image, which would hide the main message of the figure. We therefore excluded this option at an earlier stage of manuscript preparation. Instead, the values of external field and temperature are provided explicitly in the figures in the Extended Data. We have improved the caption to Fig. 2 to reflect this point.

With regard to the lowermost row of images, we have added the following sentences to the main text:

“For instance, the hopfion ring shown in the bottom row has a triangular shape at low field. With increasing magnetic field, it adopts a pentagonal and then circular shape.”

7) For figure 6, the last column with different opacity is also not terribly useful. What was the purpose of showing this?

Response: We appreciate this constructive criticism. In the revised version of the manuscript, we have replaced this panel with more informative images, which show hopfion rings calculated for bulk systems. We have also explicitly indicated the skyrmion-hopfion charges of the magnetic texture, where it is applicable. They are discussed in a new paragraph, which is devoted to topological classification of the novel magnetic textures.

8) The formatting of some references is not correct. For e.g. reference 32 is missing the title.

Response: We are grateful to the reviewer for their careful reading of the manuscript. The formatting of the references has been revised and updated.

Reviewer #2 (Remarks to the Author):

The authors Zheng et al. have submitted a highly interesting manuscript in which they analyze the stability of magnetic hopfion rings. The topic is relevant and the experimental and theoretical data is of excellent quality. Therefore, the manuscript should be published in a high-impact journal.

Response: We appreciate the highly positive evaluation of our work.

However, currently, I doubt that the study deserves publication in Nature because of a lack of significance. My main concern is the following:

1. This is not the first observation of hopfions. As the authors correctly write in the introduction, magnetic hopfions have already been observed in Ref. 24. The authors distinguish the hopfions in confined geometries, as presented in that reference, from their observation of hopfions that exist as rings around one or multiple skyrmion strings. The authors label them 'hopfion rings'. I understand that the stabilizing mechanism is different but why is that of crucial importance? Fundamentally, I do not see a higher significance of hopfion rings, especially since they can only exist around skyrmion strings. In terms of applications, I do not see that great of an advantage either. For example, in terms of current-driven motion, hopfions have been predicted to not suffer from the skyrmion Hall effect. However, in combination with the skyrmion string(s) the skyrmion Hall effect should be present. I am looking forward to read the authors' reply.

Response: We provide a detailed response about the importance of our discovery, in comparison to Ref. 24, in the next item below.

With regard to applications, we note that our group does fundamental research and applied physics is not our expertise. Following the suggestion of Reviewer #3, we have therefore excluded any mention of possible applications from the abstract.

We agree that some aspects of our findings may require additional emphasis. In particular, we would like to highlight the distinct properties of hopfion rings from the point of view of topology and homotopy groups. We believe that intriguing aspects of the topology of these solutions, which we have included in the revised version of the manuscript, are more valuable than speculations about possible applications.

With regard to the work of Kent et al., we provide extended comments below.

Please elaborate on the drastic improvement of hopfion strings over conventional hopfions that justifies a publication in Nature. After all, Ref. 24 was published in Nature Communications, so why should this study be published in Nature?

Response: There are two important aspects that distinguish our work from the seminal work of Kent et al.:

- We report the observation of hopfions in a crystal, while Kent et al. observe their hopfion in a specially synthesized system. We did not emphasize this aspect in our initial submission. We have corrected the manuscript by adding corresponding statements in the abstract and the main text.
- Our study focuses on the investigation of 3D topological magnetic solitons - localized magnetic configurations that possess the properties of ordinary particles, enabling them to move and interact with each other and their environment. It is important to note that the texture studied by Kent et al. does not meet the criteria of a soliton in this sense. The hopfion described by Kent et al. can be described as an imprinting of a Hopf fibration in a specially designed and shaped magnetic heterostructure.

Our results also have many intriguing consequences that do not overlap with the work described in Ref. 24, including:

- In the revised version of our manuscript, we provide rigorous homotopy group analysis, which introduces a new approach for the topological classification of 3D magnetic solitons with *skyrmion-hopfion topological charge*.

- In addition to the experimental observation of a wide diversity of hopfion configurations in a plate of particular thickness, we describe a broad family of hopfions that can be found in bulk and extended samples.
- We report on the hopfion zero mode, which corresponds to the screw-like motion of a hopfion along a skyrmion string.

In summary, our work not only provides direct experimental evidence that hopfions exist in crystals, but it also suggests new directions for further research. We strongly believe that the originality, novelty and significance of our work meets the high publication standards of *Nature*.

Further minor comments are as follows:

2. The authors mention a damaged layer in the experiment as well as the simulation. Is it possible to stabilize the hopfions without that damaged layer? How can we understand the importance of the damaged layer? Can you somehow 'quantify' the damaged layer so that other groups will be able to reproduce the results in experiments?

Response: Yes, it is possible to stabilize the hopfions without the damaged layer:

- In the Methods section, we wrote: "It should be noted that the presence or absence of a damaged layer in our simulations has almost no effect on the stability of the solutions shown in Fig. 4. The contrast in theoretical Lorentz TEM images in Fig. 3 also does not change significantly when the presence of a damaged layer is ignored. "
- The damaged layer is always present on the surfaces of samples prepared using focused ion beam (FIB) milling. Additional polishing of the sample surface is typically performed at a low FIB current to minimize the damaged layer thickness. Our samples were cleaned thoroughly, resulting in a minimal damaged layer thickness of approximately 5 to 10 nm. A more precise estimate of the damaged layer thickness is challenging. It is important to note that no additional sample preparation, apart from standard FIB preparation, is required to reproduce our results, so long as the sample size is approximately the same as that in our experiments. The five supplementary videos in the revised version of the manuscript can be used to guide researchers through the entire process of hopfion ring creation. We have added corresponding descriptions to the main text of the manuscript, as well as to the captions of the supplementary videos.

- In the revised version of the manuscript, we have also added simulations for the bulk model, which confirm that hopfion rings are stable without any boundary or shape effects, including damaged layers.

3. The authors strongly distinguish their hopfion rings from the hopfions observed in confined geometries, as discussed in issue 1. However, later in the paper they write that the confined geometry is important for the generation of the hopfion rings (but not for the stability). I am not sure if I understand the authors correctly but, in my understanding, this should eliminate all advantages of hopfion strings versus the hopfions in confined geometries because without a confined geometry you are not able to nucleate the hopfion rings in the first place.

Response: The hopfions that we describe do not require confinement for their stability or nucleation:

- In the revised version of the manuscript, we emphasize the fact that hopfion rings are stable in bulk systems, without needing to consider the influence of boundary or shape effects, including damaged layers.
- Confinement is always present in a real system. Although the approach used for the nucleation of these solitons in our current work relies on geometrical confinement, we anticipate that efficient techniques for their generation in bulk systems will be developed in the future. We also anticipate that hopfions may appear spontaneously (e.g., by quenching).

4. If a hopfion ring forms around two or more skyrmion strings, the strings are braided. Is the braiding essential for the stabilizing mechanism or could the hopfion ring form around straight skyrmion strings as well?

Response: The stability of hopfion rings does not rely on braiding of the skyrmion strings, and hopfion rings can form around straight skyrmion strings. This point is evident from the fact that we observe hopfion rings even around single skyrmion strings.

It is worth noting that skyrmion braiding can contribute to the Hopf charge of a magnetic texture. In the revised version of the manuscript, we have included an extended discussion of this effect, along with references to earlier and recent papers that explore this aspect:

“Figures 4e-h and Extended Data Figs 10e-h illustrate a wide diversity of stable solutions for hopfion rings in bulk systems (see Methods). Based on the general

principles of classical field theory, it is understood that the Hopf charge of skyrmion strings can be affected by longitudinal twists of skyrmions, as well as by skyrmion braiding^{36–38}. However, due to the chiral nature of the DMI present in the system studied here, stable states with skyrmion twists of multiples of 2π are not observed. Nevertheless, we speculate that such states may be possible in systems that have frustrated exchange interactions^{9,39}. On the other hand, the phenomenon of skyrmion braiding has already been demonstrated in chiral magnets²⁰. Extended Data Fig. 10h shows an example of a skyrmion braid with two hopfion rings and $H = 12$. This example can be compared with Fig. 4h, which shows straight skyrmion strings surrounded by two hopfion rings and $H = 10$.”

5. In Fig. 3 a hopfion ring around a single skyrmion string is labeled an ‘exotic state’. From a naïve approach this state should be the simplest version of a hopfion ring. Why is it ‘exotic’? Is the probability for the nucleation of this type of hopfion ring smaller than for other types?

Response: Following the protocol discussed in the manuscript, we usually obtain hopfion rings around several skyrmion strings, as illustrated in Fig. 3. The hopfion ring around a single skyrmion string appears less often in our experiment. For this reason, we described this particular configuration as an “exotic state”. We have added the following note to the text:

“Figure 3 shows exotic states with negative and positive topological charges obtained using the above protocol in a 180-nm-thick sample. **Magnetic textures with such contrast in our experiment are observed more seldom than those depicted in Fig. 3.**”

In the revised version of the manuscript, we provide five supplementary videos, which illustrate the *in situ* process of hopfion nucleation in our experimental setup.

6. About the probabilistic character of the nucleation: I think the study would benefit from quantifying the probability to nucleate a hopfion ring. What are the chances that a hopfion ring is nucleated per alternation of the magnetic field direction? Are there ways to increase this percentage? How does it depend on the number of skyrmions inside of the hopfion? How does it depend on temperature?

Response: The quantification of a hopfion ring nucleation protocol is only informative when the efficiencies of several methods are compared. We do not suggest that the protocol presented in our work is optimal or unique. The efficiency and reliability of the protocol are not the main topics of our study, which focuses on the discovery of a novel type of 3D topological magnetic soliton. The details of the protocol are provided to

ensure the transparency and reproducibility of our results. As we mention above, in the revised version of the manuscript we have provided a series of videos that illustrates the process of hopfion ring nucleation in detail. We believe that the publication of our work will stimulate research in this field, and that alternative approaches for hopfion ring nucleation will be found.

As we wrote above, we observe a hopfion ring on a single skyrmion string less often than states with more skyrmions. This fact may reflect the peculiarities of our protocol, rather than the stability of these states. In general, the number of skyrmions does not contribute significantly to the probability of hopfion ring nucleation. This point is also illustrated in the videos provided in the revised version of the manuscript.

The role of temperature is discussed in detail in the original version of the manuscript. Please see the paragraph starting with the following sentence:

“The above protocol becomes more reliable at higher sample temperature, indicating the **significant** role of thermal fluctuations for hopfion ring nucleation.”

7. Concerning the last point: The authors write that the protocol becomes more reliable at higher temperatures and that it requires more field swapping cycles to become stable at lower temperatures. How can we understand this?

Response: In the paragraph in which we discussed the influence of temperature on hopfion ring nucleation, we wrote:

“Below 180 K, nucleation of hopfion rings is still possible, but typically requires more field-swapping cycles. At lower temperature, the edge modulations can move toward the edges of the sample and disappear. In contrast, at higher temperatures the edge modulations can contract towards the center of the sample.”

We believe that this description is clear. However, we also accept that it is better to show it visually. In the revised version of the manuscript, Supplementary Videos 1-5 illustrate the entire process of hopfion ring nucleation in detail. They show that the field swapping cycles induce additional edge modulations around the perimeter of the sample. At higher temperature (180 K), such modulations often form closed loops and propagate towards the center of the sample. At lower temperature, they seldom form closed loops around the perimeter of the sample.

8. Please clarify whether or not a hopfion ring always remains stable if the magnetic field orientation is continued to be alternated after a hopfion ring has successfully been

nucleated. This would help to assure that a hopfion ring forms with a high probability in potential devices because you could simply change a field direction many times.

Response: A hopfion ring is stable in a specific range of external magnetic fields. When the magnetic field is reduced, skyrmions inside the hopfion ring experience an elliptic instability. The probability of obtaining the same hopfion ring when the field is increased remains finite, but it depends on how much the magnetic field was reduced. Supplementary Video 4 illustrates this effect. After hopfion ring nucleation, the field is reduced below the threshold value for the elliptic instability. When the field is then increased, the same hopfion ring appears in a slightly different place of the domain (in the corner). Supplementary Video 4 ends with an increase of the field in a positive direction, ultimately leading to hopfion ring collapse.

In the revised version of the manuscript, a detailed discussion of the supplementary videos has been included in the “Magnetic imaging in the TEM” part of the Methods section.

9. The authors write that Fig. 2 shows that the symmetry changes reversibly. What exactly does that mean? The process in the top row is certainly not reversible.

Response: For clarity, we have corrected the corresponding paragraph as follows:

“Figure 2 shows that the symmetry of the magnetic texture of skyrmions and hopfion rings also changes with increasing applied field. For instance, the hopfion ring shown in the bottom row has a triangular shape at the low field. With increasing magnetic field, it adopts a pentagonal and then circular shape. Such symmetry transitions are found to be reversible with respect to increasing and decreasing fields.”

10. Out of interest: Do you think it is possible to better analyze these objects by three-dimensional techniques such as holography? If not, what are the problems?

Response: By default, electron holography provides only a two-dimensional projection of a magnetization field and not a three-dimensional reconstruction. The phase shift image of a hopfion ring shown in Fig. 1g was recorded using electron holography.

In combination with tomography, *i.e.*, by recording holograms at different simple tilt angles, three-dimensional mapping of magnetic fields is in principle possible. Such a three-dimensional technique can be referred to as electron holographic tomography.

However, in its present form this technique has several limitations. The major issue is that the magnetic solitons that are studied in our work are stable only in the presence of

an external magnetic field, which is applied perpendicular to the plane of the thin TEM sample. In our TEM, the conventional objective lens is used to apply an external magnetic field to the specimen. The direction of this external magnetic field is therefore fixed in the direction of the electron beam, and it is not possible to tilt the magnetic field together with the specimen.

When the sample is tilted with respect to the electron beam, the change in the relative direction between the specimen and the external magnetic field affects the magnetic texture of the soliton. Two different states are then imaged at two different sample tilt angles. Since the stability of a hopfion ring requires a fixed direction of the external magnetic field relative to the sample, the hopfion ring moves and becomes unstable when the sample tilt is changed. In our experiments, the hopfion ring would be attracted to the edge of the sample and collapse.

Although a magnetizing holder or stage could in principle be used to tilt the field together with the specimen, such a solution is not yet readily available, especially in combination with cooling of the sample.

Reviewer #3 (Remarks to the Author):

In their paper „Hopfion rings in a cuboc chiral magnet“ the authors Fengshan Zheng et al. present evidence for the existence of so-called Hopfions using Lorentz Transmission Electron Microscopy (LTEM) and electron holography compared to micromagnetic simulations. Hopfions are three dimensional solitons predicted to be present in three dimensional chiral magnets. In contrast to skyrmion strings which (ideally) penetrate the whole specimen, Hopfions are truly localized in space, detached from sample boundaries and thus are discussed as topologically protected information carriers. The authors provide a special magnetic field protocol applied to thin FeGe platelets that allows controlled creation of stable magnetic structures with different topological comprising of Hopfions (essentially a closed string) wrapped like a belt around skyrmion tubes with various topological numbers. While Hopfions in magnetic thin film materials might have been detected previously in ref 24 (even though in this paper the evidence is not very strong) this is the first observation of what the authors call Hopfion rings in cubic chiral magnets. This is a beautiful paper and I favour publication in Nature.

Response: We appreciate the highly positive evaluation of our work.

I would like to ask the authors to comment on the following minor issues:

In the abstract potential applications are mentioned, but the authors do not expand on this. I suggest removing this half sentence or expanding the discussion concerning applications in the main text.

Response: We agree with the reviewer. Although we are able to suggest several concepts for the potential application of hopfion rings, our paper belongs to fundamental research rather than applied physics. We have therefore removed any mention of possible practical applications from the abstract.

In the caption of Fig. 1 please correct the dimensions of the platelet and please add information concerning the sample temperature.

Response: We thank the reviewer for their thorough review of the manuscript. We acknowledge that there was a typo in the figure caption, which was initially written as "size of $1\ \mu\text{m} \times a1\ \mu\text{m}$ ". We have corrected it to read "size of $1\ \mu\text{m} \times 1\ \mu\text{m}$." We have also added information about the sample temperature in the figure caption.

Please provide more details concerning the sample preparation using FIB and the damage layer and move this part to the main text, this seems to be important information.

Response: Although the sample was prepared using a standard procedure, we acknowledge the significance of the damaged layer. Due to length limitations, we cannot incorporate the entire paragraph that discusses the damaged layer in the main text. Instead, we have added an extra note to draw the readers' attention to the relevant part of the Methods section:

"We also took into account the presence of a thin damaged layer on the sample surface (Fig. 1a), which typically results from sample preparation by focused ion beam milling. (See the Methods section for more details about micromagnetic calculations, **the properties of the damaged layer**, and Lorentz image simulations)."

Chiral bobbbers are mention e.g. in the caption of Fig. 2. Could the authors comment on their stability? Do they occur at defects, or why are they stable?

Response: Chiral bobbbers are statically stable magnetic solitons. They are stable in a certain range of fields, and can either appear as isolated entities or interact with other solitons, including skyrmions strings. Such coupled states, in which chiral bobbbers are coupled to one or a few skyrmion strings, were observed experimentally in Ref. 31 (Zheng et al., Nature Nanotechnology 13, 451-455, 2018).

On page 3 right side bottom. The authors mention that intermediate configurations are observed seldomly. What does seldom mean? How many instances have been observed in how many experiments in total?

Response: "Seldom" means that we did not observe these intermediate configurations frequently in our experiments and cannot provide many images of them. These intermediate states are not statically stable configurations and typically only appear dynamically. In the revised version of the manuscript, we have included five movies that illustrate the *in situ* nucleation of hopfions in our experimental setup. We believe that these videos illustrate all of the details of the hopfion nucleation process.

At the bottom of page three the authors mention that thermal fluctuations have an influence on the nucleation induced by the magnetic field protocol. Can this be corroborated in temperature dependent simulations (see also Fig.3 for simulations of images taken at different temperatures)?

Response: As we wrote in our reply to Reviewer #2 above, the nucleation process presented in our work is probably not unique, and we expect that more efficient methods will be proposed in the future.

With regard to temperature-dependent simulations:

- Our micromagnetic model includes thermal effects to an extent *via* the material parameters. In particular, our parameters are adopted for $T = 95$ K, which is a typical temperature for a TEM experiment performed using liquid nitrogen. The adoption of these parameters was previously established in the work of Zheng et al. (Nature Nanotechnology 13, 451-455, 2018). The results presented in our study illustrate the predictive power of our model and adopted parameters.
- For higher temperatures, the role of thermal fluctuations increases significantly. Our sample comprises approximately 10^9 magnetic atoms, whose dipole-dipole interactions are important for the energy balance. Unfortunately, accurate calculations that include temperature (in an atomistic model) for this number of atoms are not possible using current computers.

Did the authors use different platelet sizes (in-plane size)? Does this have an influence on nucleation?

Response: We attempted to reproduce the hopfion structure in a larger sample, which had a lateral dimension of $3 \mu\text{m}$ and a thickness of 180 nm . However, we encountered difficulties in forming a closed loop of edge modulations for such a larger sample. The

primary challenge resulted from limitations of FIB technique, as it is extremely challenging to achieve a uniform thickness across a larger sample. Conversely, when working with smaller sample sizes, we observed that there was insufficient space for a hopfion to stabilize due to edge effects. Taking these practical considerations into account, we chose a lateral dimension of approximately 1 μm for our experiments.

Can the authors move the composite particles with tilted magnetic field? The authors mention that tilt angles should not exceed 5 degrees for nucleation (this information might also be moved to the main text if space allows), but what happens if the particle has been nucleated and the field is tilted?

Response: An additional tilt of the specimen will introduce an in-plane component of the applied field, as the direction of the external magnetic field is fixed to the direction of the electron beam (the z direction). The hopfion will then be guided by the in-plane field and attracted to the sample edges.

In the figure shown below, we provide a series of images that shows this effect when the magnetic field is reduced after the nucleation of a hopfion ring (left: 256 mT; middle and right: 144 mT). It can be seen that the hopfion ring is then attracted to the edge and moves along it towards the corner.

In terms of the nucleation process illustrated in Supplementary Videos 1-5, edge modulations would occur and propagate from only one edge of the sample in a tilted external magnetic field, and closed loops would not be formed.

Following the reviewer suggestion, we have added the following note in the main text:

“The tilt angle of the external magnetic field to the plate normal is an essential parameter for hopfion ring nucleation. In our experiment, we found that the tilt angle of the field should not exceed 5 degrees. Otherwise, the edge modulations primarily form on one side of the sample, resulting in a strongly asymmetric configuration.”

In Fig. S1a a section of a ring is visible with slightly higher contrast at the ends, why is this so? Can rings rip? Or is this section pinned by defects?

Response: In low magnetic fields, clusters of skyrmions exhibit skyrmion braiding (to a small extent). In the figure mentioned by Reviewer #3 and reproduced below, in a field of 165 mT the skyrmion exhibits braiding with the hopfion ring next to it.

The figure shows that the presence of this skyrmion becomes more evident with increasing magnetic field, as the skyrmion becomes stable and moves to the edge and then to the corner.

Reviewer Reports on the First Revision:

Referees' comments:

Referee #1 (Remarks to the Author):

In the revised manuscript the authors have satisfactorily addressed the comments raised by all reviewers. I appreciate the efforts taken by the authors to complete additional experiments and analysis. They have added more information about how Hopfion rings will interact with their surrounding and behave under external fields, which is unique. Overall the paper now meets the novelty and uniqueness criterion required by Nat. Only two minor comments –

- 1) At some places in the text, the spin textures are referred to as Hopfions vs. Hopfion rings. This should be kept consistent and perhaps referred to as Hopfion rings through out the text.
- 2) Since the authors have included a phase shift image of the Hopfion ring in Figure 1, for completeness sake, it would be good to include the comparison of the phase shift to show the difference between the phase shift expected from a skyrmion bag/Bloch domain wall that extends through the thickness and the Hopfion ring. This was clearly shown in the paper on chiral bobbers which helps clarify the difference quantitatively between the two structures rather than qualitatively saying “strong contrast” vs. “weak contrast”.

Referee #2 (Remarks to the Author):

The authors Zheng et al. have resubmitted their significantly improved manuscript and I appreciate their replies to my comments.

In my initial report I had doubts about the novelty of the presented results which was the main reason why I had refrained from recommending this manuscript for publication in Nature. This issue has been resolved by the authors. I agree that stabilizing a whole variety of hopfion rings in crystals is different from stabilizing a single hopfion in a specially synthesized material. Furthermore, I appreciate the additional theoretical homotopy group analysis of the observed textures. It definitely adds to the manuscript by distinguishing it further from the existing literature and it suggests new directions for hopfion-related research. For these reasons, I can now recommend this paper for publication in Nature.

I have also read the replies to the other referee reports and want to say that I find the additional figure that shows that the hopfion rings move when the field is tilted extremely interesting (reply to referee 3). This result underlines the significance of ‘free’ hopfion rings versus the confined hopfions that have been observed previously. These specific results are of course still premature but I would definitely appreciate a follow-up study on this effect.

The authors have replied convincingly to most of the other issues that I had brought up. I do not need to see the paper again but I would like to highlight a few misunderstandings of my previous

comments, in case the authors would like to further improve upon them:

6.) To me, the probabilistic character of the nucleation process is still not completely clear. The authors refrain from analyzing the probabilities for nucleating specific types of hopfion rings because better protocols may exist that would yield different probabilities or would even have a non-probabilistic nature. Instead, the authors provide interesting videos of several nucleation events. The videos help understanding the nucleation but still, analyzing the probabilities might be a worthwhile effort for a follow-up study.

7.) I was asking about the temperature dependence and the authors replied: "Below 180 K, nucleation of hopfion rings is still possible, but typically requires more field-swapping cycles. At lower temperature, the edge modulations can move toward the edges of the sample and disappear. In contrast, at higher temperatures the edge modulations can contract towards the center of the sample. We believe that this description is clear." Indeed, the *description* is clear but what is the *explanation* for this behavior?

8.) With my question I had something else in mind: The authors explain that they alternate the magnetic field until a hopfion ring is nucleated. I was wondering what happens if they continue to alternate the magnetic field once a hopfion ring is stable. Will the hopfion ring remain stable 'forever' or is there a possibility for annihilation due to the probabilistic nature of the process? If the first was true, this would mean that one could reliably create hopfions without looking at the magnetic texture because one could simply alternate the magnetic field many times. One could easily estimate the required number of alternations if one would know about the probability of nucleation per alternation of the field. This is why I brought up issue 6.

One last comment: I understand why H could not be defined for the textures in Fig. 4b,c but I think it should be mentioned in the figure caption as well. Otherwise the labels (-3,_) and (-2,_) are confusing.

Referee #3 (Remarks to the Author):

The authors have responded to the questions and comments adequately and I would (as already mentioned in the first report) favor publication.

Author Rebuttals to First Revision:

Below, we provide a point-by-point response to the reviewers' questions and comments.

Reviewer #1 (Remarks to the Author):

In the revised manuscript the authors have satisfactorily addressed the comments raised by all reviewers. I appreciate the efforts taken by the authors to complete additional experiments and analysis. They have added more information about how Hopfion rings will interact with their surrounding and behave under external fields, which is unique. Overall the paper now meets the novelty and uniqueness criterion required by Nat.

Response: We are very grateful for the referee's constructive comments and suggestions, which have significantly improved the paper, as well as for the highly positive evaluation of our work and for the recommendation for publication in *Nature*.

Only two minor comments –

1) At some places in the text, the spin textures are referred to as Hopfions vs. Hopfion rings. This should be kept consistent and perhaps referred to as Hopfion rings throughout the text.

Response: Following the referee's recommendation, in the final version of the manuscript we have replaced the term *hopfion* with *hopfion ring* when referring to our spin textures.

2) Since the authors have included a phase shift image of the Hopfion ring in Figure 1, for completeness sake, it would be good to include the comparison of the phase shift to show the difference between the phase shift expected from a skyrmion bag/Bloch domain wall that extends through the thickness and the Hopfion ring. This was clearly shown in the paper on chiral bobbers which helps clarify the difference quantitatively between the two structures rather than qualitatively saying "strong contrast" vs. "weak contrast".

Response: In the new Extended Data Fig. 10, we have provided a direct comparison between experimental and theoretical phase shift images. In order to further support this point, we have improved Extended Data Fig. 9 (previously Extended Data Fig. 7) to show phase shift images for both a hopfion ring and a skyrmion bag/domain wall as a function of specimen thickness. We have also included an additional hopfion ring configuration. A reference to the new figure in the Extended Data has been added to the main text.

Reviewer #2 (Remarks to the Author):

The authors Zheng et al. have resubmitted their significantly improved manuscript and I appreciate their replies to my comments.

In my initial report I had doubts about the novelty of the presented results which was the main reason why I had refrained from recommending this manuscript for publication in *Nature*. This issue has been resolved by the authors. I agree that stabilizing a whole variety of hopfion rings in crystals is different from stabilizing a single hopfion in a specially synthesized material. Furthermore, I appreciate the additional theoretical homotopy group analysis of the observed textures. It definitely adds to the manuscript by distinguishing it further from the existing literature and it suggests new directions for hopfion-related research. For these reasons, I can now recommend this paper for publication in *Nature*.

Response: We sincerely appreciate the referee's positive comments about our work and for the recommendation for publication in *Nature*.

I have also read the replies to the other referee reports and want to say that I find the additional figure that shows that the hopfion rings move when the field is tilted extremely interesting (reply to referee 3). This result underlines the significance of 'free' hopfion rings versus the confined hopfions that have been observed previously. These specific results are of course still premature but I would definitely appreciate a follow-up study on this effect.

Response: We are delighted that the reviewer shares our enthusiasm for the results presented in our work. The zero mode of the hopfion ring on a skyrmion string is indeed an intriguing phenomenon, which deserves special attention. We plan to explore this effect in detail in future papers, both in chiral magnets and in other systems.

The authors have replied convincingly to most of the other issues that I had brought up. I do not need to see the paper again but I would like to highlight a few misunderstandings of my previous comments, in case the authors would like to further improve upon them:

Response: We apologize for misunderstanding some of the referee's comments. We hope that the following responses to these comments are convincing.

6.) To me, the probabilistic character of the nucleation process is still not completely clear. The authors refrain from analyzing the probabilities for nucleating specific types of hopfion rings because better protocols may exist that would yield different probabilities or would even have a non-probabilistic nature. Instead, the authors provide interesting

videos of several nucleation events. The videos help understanding the nucleation but still, analyzing the probabilities might be a worthwhile effort for a follow-up study.

Response: In order to address the probabilistic nature of the hopfion ring nucleation process in detail, it is necessary to eliminate the influence of the operator of the electron microscope on the results. In the present experiments, the external magnetic field was controlled manually by an operator in response to observations of Lorentz image contrast. In the future, unbiased probability analysis can be achieved by making use of automation of the field-swapping protocol, as well as through the collection of larger datasets. However, these tasks go beyond the scope of the present work.

7.) I was asking about the temperature dependence and the authors replied: “Below 180 K, nucleation of hopfion rings is still possible, but typically requires more field-swapping cycles. At lower temperature, the edge modulations can move toward the edges of the sample and disappear. In contrast, at higher temperatures the edge modulations can contract towards the center of the sample. We believe that this description is clear.” Indeed, the *description* is clear but what is the *explanation* for this behavior?

Response: This behavior is attributed to the energy landscape and energy barriers. The explanations are provided in the following text of the same paragraph (marked in bold):

“At lower temperature, the edge modulations can move toward the edges of the sample and disappear. In contrast, at higher temperatures the edge modulations can contract towards the center of the sample. Above 200 K, abrupt contraction of the edge modulations can lead to their collapse. **The behavior of the edge modulations at different temperatures can be explained by the presence of an energy barrier that prevents their contraction towards the center of the domain. The probability of overcoming this energy barrier then becomes greater at high temperature.**”

8.) With my question I had something else in mind: The authors explain that they alternate the magnetic field until a hopfion ring is nucleated. I was wondering what happens if they continue to alternate the magnetic field once a hopfion ring is stable. Will the hopfion ring remain stable ‘forever’ or is there a possibility for annihilation due to the probabilistic nature of the process? If the first was true, this would mean that one could reliably create hopfions without looking at the magnetic texture because one could simply alternate the magnetic field many times. One could easily estimate the required number of alternations if one would know about the probability of nucleation per alternation of the field. This is why I brought up issue 6.

Response: States with hopfion rings are metastable, and are protected by finite energy barriers. These solutions remain stable in the presence of small perturbations. However, in principle the system's energy can be reduced if a hopfion ring collapses. Therefore, application of the field swapping protocol may lead to either nucleation or collapse of the hopfion ring. In order to clarify this point, we have improved the following paragraph and added a note (see the text marked in red):

“Remarkably, the sequential application of the above field swapping protocol can result in the nucleation of a second hopfion ring with high probability. Figure 1g shows an example of such a double hopfion ring around three skyrmion strings. Further images with multiple hopfion rings are provided in Extended Data Figs 5 and 6. **These images illustrate the successful contraction of the second ring. However, because hopfion rings are metastable states, further application of the field swapping protocol can cause a transition into a lower energy state, resulting in collapse of the hopfion rings.**”

One last comment: I understand why H could not be defined for the textures in Fig. 4b,c but I think it should be mentioned in the figure caption as well. Otherwise the labels (-3,_) and (-2,_) are confusing.

Response: In order to avoid any confusion, in the revised version of the manuscript we have corrected Fig. 4 and Extended Data Fig. 8 (previously Extended Data Fig. 10). Wherever there is uncertainty in the calculation of the Hopf index, instead of the symbol “_” we have added “unc.”. We have also added the following note to the figure caption:

“Note that, for the states shown in b and c, the Hopf charge is uncertain. (See the main text for details).”

Reviewer #3 (Remarks to the Author):

The authors have responded to the questions and comments adequately and I would (as already mentioned in the first report) favor publication.

Response: We would like to thank the reviewer for the highly positive evaluation of our work and for supporting its publication in *Nature*.